# The complete sequence and comparative analysis of ape sex chromosomes

Apes possess two sex chromosomes—the male-specific Y chromosome and the X chromosome, which is present in both males and females. The Y chromosome is crucial for male reproduction, with deletions being linked to infertility[1]. The X chromosome is vital for reproduction and cognition[2]. Variation in mating patterns and brain function among apes suggests corresponding differences in their sex chromosomes. However, owing to their repetitive nature and incomplete reference assemblies, ape sex chromosomes have been challenging to study. Here, using the methodology developed for the telomere-to-telomere (T2T) human genome, we produced gapless assemblies of the X and Y chromosomes for five great apes (bonobo (*Pan paniscus*), chimpanzee (*Pan troglodytes*), western lowland gorilla (*Gorilla gorilla gorilla*), Bornean orangutan (*Pongo pygmaeus*) and Sumatran orangutan (*Pongo abelii*)) and a lesser ape (the siamang gibbon (*Symphalangus syndactylus*)), and untangled the intricacies of their evolution. Compared with the X chromosomes, the ape Y chromosomes vary greatly in size and have low alignability and high levels of structural rearrangements—owing to the accumulation of lineage-specific ampliconic regions, palindromes, transposable elements and satellites. Many Y chromosome genes expand in multi-copy families and some evolve under purifying selection. Thus, the Y chromosome exhibits dynamic evolution, whereas the X chromosome is more stable. Mapping short-read sequencing data to these assemblies revealed diversity and selection patterns on sex chromosomes of more than 100 individual great apes. These reference assemblies are expected to inform human evolution and conservation genetics of non-human apes, all of which are endangered species.

Therian X and Y chromosomes are thought to have originated from a pair of autosomes around 170 million years ago[3]. The X chromosome, which is typically present as two copies in females and one copy in males, has mostly retained the gene content and order from the original autosomal pair[4]. The Y chromosome, which is typically present as one copy in males, has acquired the sex-determining gene *SRY* and other male-specific genes and mutations, which were fixed by inversions that prevented recombination between the Y and X chromosomes over most of their lengths[5,6]. Lacking recombination, the Y chromosome has contracted in size and accumulated deleterious mutations and repetitive elements, leading to differences in size and gene content between the Y and X chromosomes. The recent human T2T (gapless and complete) assembly revealed an X chromosome of around 154 Mb with 796 protein-coding genes[7], and a Y chromosome of around 62 Mb with 106 protein-coding genes[8]. In addition to the pseudoautosomal regions (PARs), where the Y chromosome still recombines with the X chromosome, and ancestral regions, which originated from the original autosomal pair, the human Y chromosome has long ampliconic regions with extensive intrachromosomal homology. Ampliconic regions harbour palindromes—long inverted repeats that undergo gene conversion, which counteracts the accumulation of deleterious mutations[9]. Similar to the human Y chromosome, the human X chromosome possesses PARs[7], ancestral regions and several palindromes[10].

Whereas human sex chromosomes have recently been completely sequenced[7,8], the sex chromosomes in our closest relatives—non-human apes—remain incompletely characterized. Owing to the haploid nature and high repetitive element content of the Y chromosome, most previous studies have assembled female genomes, omitting the Y chromosome altogether[11]. Ape Y chromosomes have sometimes been sequenced with targeted methods[6,12,13] or via shotgun sequencing of male genomes[14,15], but such assemblies are usually fragmented, collapsed and incomplete. Ape X chromosomes have been deciphered to a greater level of contiguity (for example, in refs. 16–18), but their assemblies—particularly for long satellite arrays—have remained unfinished, preventing their complete characterization.

Previous cytogenetic studies demonstrated lineage-specific amplifications and rearrangements leading to large size variations among great ape Y chromosomes (for example, ref. 19). The initial assemblies of the human and chimpanzee Y chromosomes revealed remarkable differences in structure and gene content[6,12] despite short divergence time, and an acceleration of substitution rates and gene loss on the Y chromosome was observed in the common ancestor of bonobo and chimpanzee[15]. The Y chromosome of the common ancestor of great apes probably already possessed ampliconic sequences and multi-copy gene families[15], and all ape sex chromosomes share the same evolutionary strata[14] while experiencing lineage-specific

expansions and loss of ampliconic genes[14,15]. This progress notwithstanding, the lack of complete ape sex chromosome assemblies has prevented detailed inquiries into the evolution of ampliconic regions, palindromes, segmental duplications, structural variants, satellites, transposable elements and gene copy number. Here, utilizing the experimental and computational methods developed for the T2T assembly of the human genome[8,20], we deciphered the complete sequences of sex chromosomes from six ape species and studied their structure and evolution.

## Ape sex chromosome assemblies

To perform a comparative analysis of great ape sex chromosomes, we built genome assemblies for most extant great ape species—bonobo, chimpanzee, western lowland gorilla (hereafter referred to simply as gorilla), Bornean orangutan (hereafter B. orangutan) and Sumatran orangutan (hereafter S. orangutan). We also assembled the genome of an outgroup—the siamang, representing gibbons (lesser apes). The assemblies included two pairs of closely related species: B. orangutan and S. orangutan, which diverged from each other approximately 1 million years ago (Ma), and chimpanzee and bonobo, which diverged from each other around 2.5 Ma (Supplementary Table 1). The human lineage diverged from the *Pan*, gorilla, *Pongo* and gibbon lineages approximately 7, 9, 17 and 20 Ma, respectively (Fig. 1a and Supplementary Table 1). The studied species differ in their dispersal and mating patterns (Supplementary Table 2), potentially affecting sex chromosome structure and evolution. We isolated high-molecular-weight DNA from male cell lines for these species (Supplementary Fig. 1, Supplementary Table 3 and Supplementary Notes 1 and 2) and used it for high-coverage Pacific Biosciences (PacBio) HiFi, Ultra-Long Oxford Nanopore Technologies (UL-ONT) and Hi-C sequencing (see Methods). The sequencing depth among samples ranged from 54 to 109× for HiFi, 28 to 73× for UL-ONT and 30 to 78× for Hi-C (Supplementary Table 4). We had access to parental DNA for the studied bonobo and gorilla individuals (Supplementary Table 5) and sequenced it to 51–71× depth with Illumina short-read technology (Supplementary Table 4).

Genome assemblies were generated with Verkko[21] using the HiFi and UL-ONT data, with haplotypes phased using either parental *k*-mers or Hi-C evidence (Methods). The sex chromosomes were clearly distinguishable from the autosomes in the assembly graphs, with several X and Y chromosomes assembled completely with telomeres on each end (Supplementary Fig. 2). The remaining sex chromosomes were finished via manual curation and validated, resulting in version 1.1 of the assemblies (Supplementary Table 6 and Methods).

Altogether, we generated T2T assemblies for siamang and B. orangutan X and Y chromosomes, for which prior assemblies were unavailable, and for bonobo, chimpanzee, gorilla and S. orangutan X and Y chromosomes, for which lower-quality assemblies were available[12,15–18] (Fig. 2). Compared with the previous assemblies, newly generated sequences accounted for 24–45% and 2.6–16% of the total chromosome length on Y and X chromosomes, respectively (8.6–30 Mb and 3.9–28 Mb of sequence, respectively; Supplementary Table 7). The sequences gained in the T2T assemblies had a high frequency of motifs able to form non-canonical (non-B) DNA structures (Fig. 2; $P < 2.2 \times 10^{-16}$ for logistic regressions in each species with previous assemblies; Supplementary Table 8), which are known to be problematic sequencing targets[22]. Combining sequencing technologies, as done here, remedies sequencing limitations in such regions[22].

The variation in length was larger among the Y chromosomes than among the X chromosomes across the studied species (including human X and Y chromosomes[7,8]; Fig. 2). Ape Y chromosomes ranged in size from 30 Mb in siamang to 68 Mb in S. orangutan and differed by as much as 19 Mb between the two orangutan species and 11 Mb between bonobo and chimpanzee. The X chromosomes ranged in size from 154 Mb in chimpanzee and human to 178 Mb in gorilla and differed

by only 1.5 Mb between the two orangutan species and 6.3 Mb between bonobo and chimpanzee.

## High interspecific variation on the Y chromosome

Across all pairwise species comparisons, the percentage of sequence aligned was lower for Y chromosomes than for X chromosomes (Fig. 1b). Only 14–27% of the human Y chromosome was covered by alignments to the other ape Y chromosomes, whereas as much as 93–98% of the human X chromosome was covered by alignments to the other ape X chromosomes (Fig. 1b,c). The same pattern was observed for closely related species, with only 60–87% of the Y chromosome, but more than 95% of the X chromosome, aligned between them (Fig. 1c).

By analysing sequence similarity between the X and Y chromosomes of the same species, we identified PARs (Fig. 1c, Supplementary Table 9 and Methods), which undergo recombination and thus differ only at the haplotype level between the two sex chromosomes[6]. All species possessed a homologous 2.2- to 2.5-Mb PAR1, but independently acquired PAR2 sequences were identified in human and bonobo. The PAR2 is approximately 330 kb long in human[8] and approximately 95 kb in bonobo (data from this study), yet they are not homologous (Supplementary Note 3). The subsequent analyses excluded PARs unless indicated otherwise.

In the sequences with interspecies variation, 83–86% of base pairs on the X chromosome and 99% of bases on the Y chromosome were affected by large-scale structural variants (Fig. 1c, Supplementary Figs. 3 and 4), and the remaining base pairs were affected by single nucleotide variants (Supplementary Table 10 and Methods). Inversions were abundant on the Y chromosome (Supplementary Table 10), consistent with its palindromic architecture. Inversions and insertions were approximately eightfold and threefold longer on the Y chromosome than on the X chromosome, respectively (average sizes of 12.1 Mb versus 1.5 Mb and 38.2 kb versus 11.1 kb, respectively; $P < 2.2 \times 10^{-16}$, Wilcoxon ranked-sum tests). The number of structural variants correlated positively with the lengths of phylogenetic branches (Supplementary Fig. 5 and Supplementary Table 11), with a greater slope for the Y chromosome (15.8 structural variants per Mb per million years) than for the X chromosome (6.1 structural variants per Mb per million years), indicating a more rapid accumulation of structural variants on the Y chromosome than on the X chromosome. To identify structural variants with potential functional significance in the human lineage, we studied overlaps with genes for 334 and 1,711 human-specific structural variants on the Y and X chromosomes, respectively (Supplementary Data 1–5 and Supplementary Table 12). On the Y chromosome, we detected an insertion of the previously reported 3.7-Mb X-transposed region—a human-specific duplication from the X chromosome to the Y chromosome[6]—that includes 13 genes. Outside of gene copy number changes, human-specific inversions affected 11 genes on the Y chromosome, and human-specific insertions and deletions affected 23 genes on the X chromosome. Thus, structural variants represent one of the dominant types of genetic variation on the X chromosome and particularly on the Y chromosome, and might have functional consequences.

The phylogenetic analysis of multi-species alignments (Methods) for the X chromosome, and separately for the Y chromosome, revealed the expected species topology (Fig. 1a) but detected higher substitution rates on the Y chromosome than on the X chromosome for all the branches (Fig. 1d), consistent with male mutation bias[23,24]. For instance, the human–chimpanzee divergence was 2.68% on the Y chromosome and 0.97% on the X chromosome. For the Y chromosome, we detected an 11% acceleration of substitution rates in the *Pan* lineage and a 9.2% slowdown in the *Pongo* lineage, compared with substitution rates in the human lineage (significant relative rate tests; *P* values in Supplementary Table 13). For the X chromosome, substitution rates were more similar in magnitude among the branches (Supplementary Table 13). These results indicate a stronger male mutation bias for the *Pan* lineage and

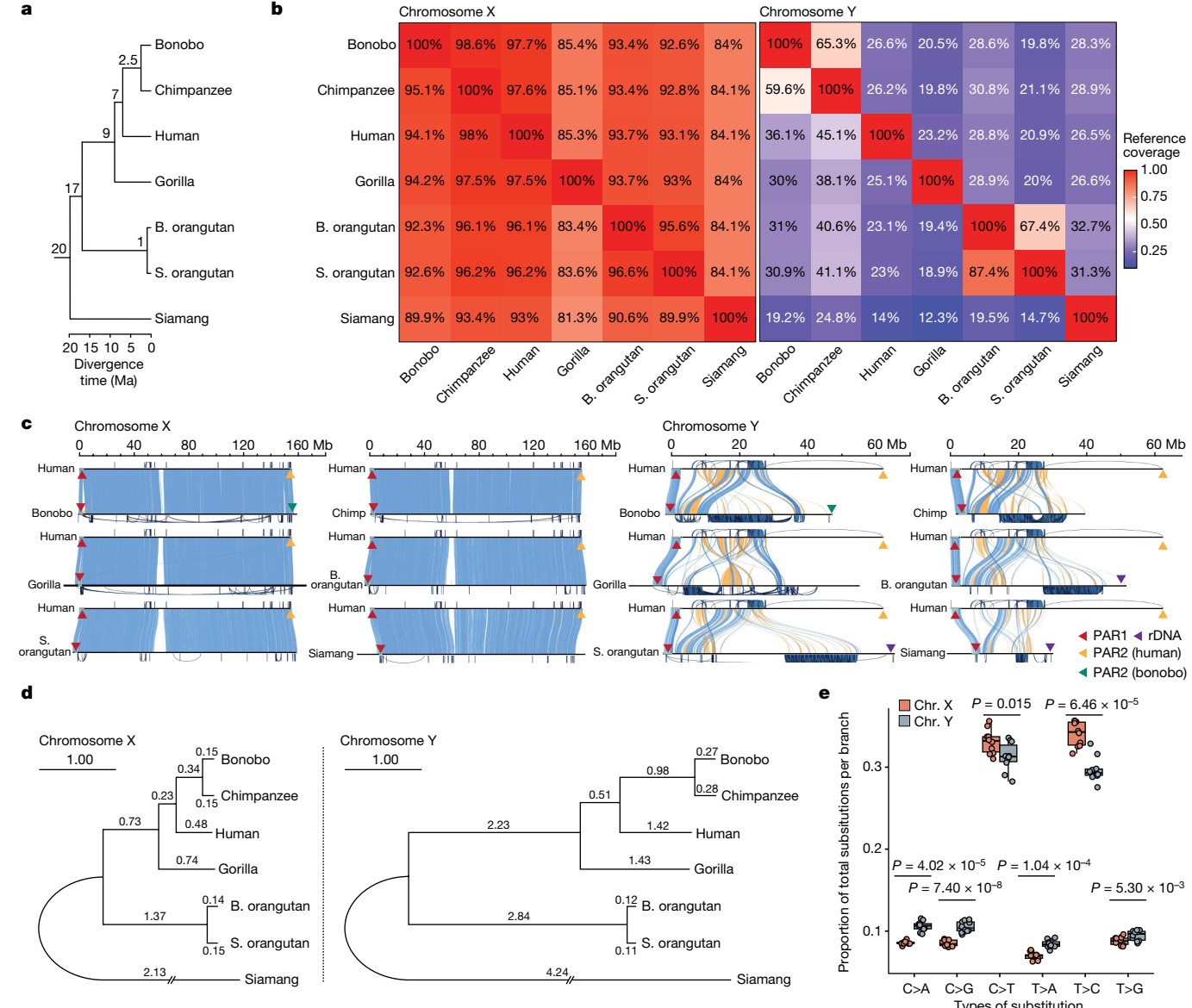

**Fig. 1 | Chromosome alignability and divergence. a**, The phylogenetic tree of the species in the study (see Supplementary Table 1 for references of divergence times). **b**, Pairwise alignment coverage of X and Y chromosomes (percentage of reference, as shown on the *x* axis, covered by the query, as shown on the *y* axis). **c**, Alignment of ape sex chromosomes against the human T2T assembly[8,20]. Blue and yellow bands indicate direct or inverted alignments, respectively. PARs and ribosomal DNA arrays (rDNA) are indicated by triangles (not to scale). Intrachromosomal segmental duplications are drawn outside the axes. The scale bars are aligned to the human chromosome. rDNA, ribosomal DNA. **d**, Phylogenetic trees of nucleotide sequences on the X and Y chromosomes[69]. Branch lengths (substitutions per 100 sites) were estimated from multi-species alignment blocks including all seven species. **e**, A comparison of the proportions of six single-base nucleotide substitution types among total nucleotide substitutions per branch between X and Y (excluding PARs). The distribution of the proportion of each substitution type across 10 phylogenetic branches is shown as a dot plot (all data points are plotted) over the box plot. Box plots show the median as the centre line and the first and third quartiles as bounds; the whiskers extend to the closer of the minimum and maximum value or 1.5 times the interquartile range. The significance of differences in means of substitution proportions between X and Y chromosomes for each substitution type was evaluated with a two-sided *t*-test on the data from all ten branches (Bonferroni correction for multiple testing was applied).

a weaker bias for the *Pongo* lineage than for the human lineage. Strong male mutation bias in the *Pan* lineage is consistent with increased sperm production due to sperm competition (Supplementary Table 2).

Comparing nucleotide substitution spectra between the two sex chromosomes, we found C>A, C>G, T>A and T>G substitutions to be significantly more abundant on the Y chromosome than on the X chromosome, and C>T and T>C substitutions to be more abundant on the X chromosome than on the Y chromosome (Fig. 1e). These findings are broadly consistent with sex-specific signatures of de novo mutations from other studies; C>A, C>G and T>G were shown to be enriched in

paternal de novo mutations, whereas C>T mutations were enriched in maternal de novo mutations[25]. C>G mutations might be related to meiotic double-strand breaks in the male germline[26].

## Ampliconic regions and palindromes

Outside of PARs, we separated the assemblies into ancestral, ampliconic and satellite regions (Fig. 2, Supplementary Table 14, Supplementary Data 2 and Methods). The ancestral regions (also called 'X-degenerate' on the Y chromosome[6]), which are the remnants of the autosomal past,

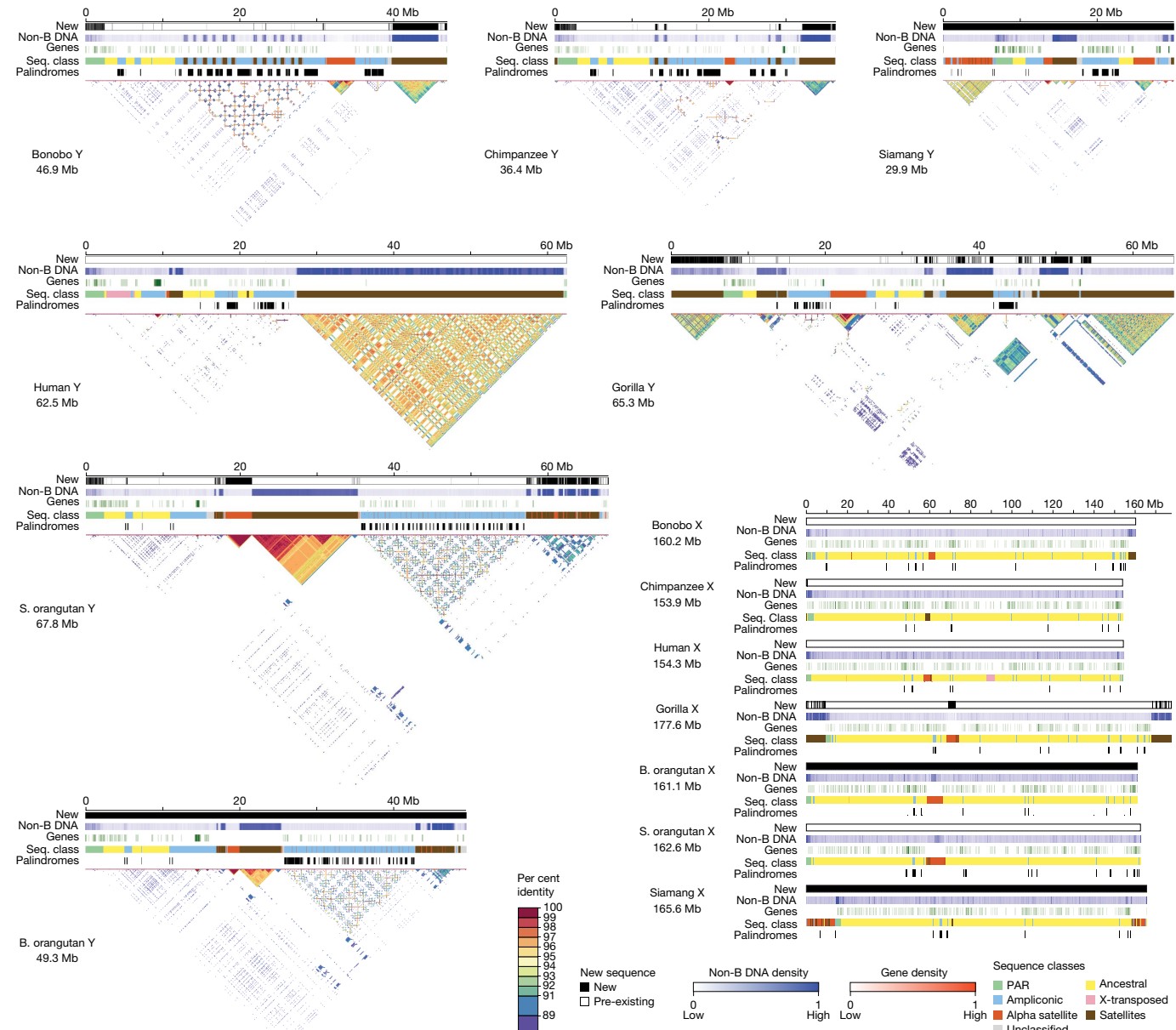

**Fig. 2 | Sequences gained, non-B-DNA, genes, sequence classes, palindromes and intrachromosomal similarity in the assemblies.** Tracks for newly generated sequence (black) relative to previous assemblies, non-B-DNA density, gene density (up to 11 genes per 100-kb window), sequence classes (seq. class) and palindromes (black). The X and Y chromosomes are portrayed on different scales. No previous references existed for the Bornean orangutan or siamang, thus the solid black bars for the new sequence tracks. No new sequence was added to the existing T2T human reference in this study and thus the new sequence tracks are empty (white). The gene density tracks are normalized across all species and chromosomes; the non-B-DNA density tracks are calibrated independently for each chromosome; in both cases, darker shades indicate higher density. Self-similarity dot plots using a modified version of Stained Glass[70] are shown for the Y chromosomes; satellite arrays are visible as blocks of colour, segmental duplications appear as horizontal lines, and inverted or palindrome repeats are shown as vertical lines.

ranged in size from 138 to 147 Mb among species on the X chromosome, but were much shorter (3.6–7.5 Mb) on the Y chromosome, consistent with sequence loss due to the lack of recombination on the Y chromosome. We did not find X-transposed regions[6] on the Y chromosomes of non-human apes (Supplementary Note 4).

Ampliconic regions, defined as long (more than 90 kb) multi-copy sequences with more than 50% identity between copies (see Methods), ranged from 3.8 to 6.9 Mb on the X chromosome, but were longer on the Y chromosome (from 9.7 to 28 Mb), and contributed substantially to variation in the length of the Y chromosome among species (Fig. 2 and Supplementary Table 14). These regions were shorter (by 2.5–25 Mb) in previous Y assemblies[12,15] than in our T2T-Y assemblies, suggesting

their collapse in the earlier assemblies. Ampliconic regions on the X chromosome were shared among species to a large degree (Extended Data Fig. 1a); for instance, we could detect their homology among the African great apes. By contrast, we could detect homology between Y chromosome ampliconic regions only in pairs of closely related species—such as between bonobo and chimpanzee, and between B. orangutan and S. orangutan (Extended Data Fig. 1b)—yet these regions still differed in organization (Supplementary Fig. 6), suggesting extremely rapid evolution.

Within ampliconic regions, we located palindromes—defined as inverted repeats, larger than 8 kb in size, of sequences with at least 98% identity (that is, arms)—frequently separated by a spacer

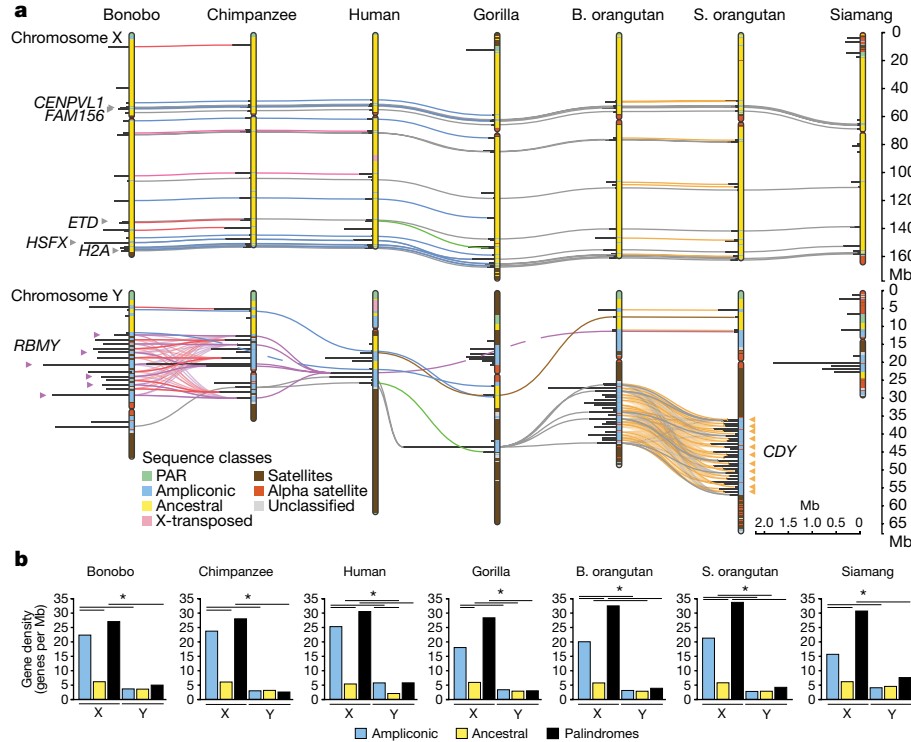

**Fig. 3 | Conservation of palindromes and gene density in different sequences classes. a**, Palindromes are shown as horizontal lines perpendicular to the chromosomes (painted with sequence classes); palindromes shared among species are connected by coloured lines (different colours are used for unique species combinations, may be dashed when horizontally passing through species without sharing, opacity reduced in regions with dense palindrome sharing). Several gene families that expanded in lineage-specific palindromes on the Y (*CDY* and *RBMY*) and that are present in palindromes shared among

species on the X chromosome (*CENPVL1*, *FAM156*, *ETD*, *HSFX* and *H2A*) are indicated. See Supplementary Tables 36, 37, 38 and 41 for the original data. **b**, Gene density for different sex chromosome sequence classes. The significance of differences in gene densities was computed using goodness of fit (chi-squared) test with Bonferroni correction for multiple tests. Asterisks indicate significant differences in gene density ($P < 0.05$). See Supplementary Table 38 for the original data and $P$ values. An interactive version of this plot can be found at https://observablehq.com/d/6e3e88a3e017ec21.

(Figs. 2 and 3a, Supplementary Data 3 and Methods). Palindromes on the Y chromosome were on average two to three times longer (Fig. 3a and Supplementary Fig. 7a; with significant $P$ values for one-sided Wilcoxon rank-sum tests in most cases (Supplementary Table 15)), and had significantly higher coverage ($P = 2.12 \times 10^{-3}$, two-sided Wilcoxon rank-sum test; Supplementary Table 15), than on the X chromosome for all species, supporting their role in rescuing deleterious mutations through intrachromosomal recombination and gene conversion on the Y chromosome[5,9]. Consistent with gene conversion, we found higher GC content in palindrome arms than spacers on both X and Y chromosomes ($P = 3.08 \times 10^{-2}$ and $P = 1.04 \times 10^{-2}$, respectively, two-sample one-sided $t$-tests; Supplementary Fig. 7b). Palindromes on the X chromosome were conserved among species (Fig. 3a and Supplementary Table 16); 21, 12 and 9 homologous palindrome clusters were shared among African great apes, among all great apes and among all species analysed, respectively. Palindromes on the Y chromosome were substantially less conserved (Fig. 3a and Supplementary Table 16); two, one and no homologous palindrome clusters were shared among African great apes, among all great apes and among all species analysed, respectively. Y palindromes were frequently species-specific or shared by closely related species only.

Segmental duplications—defined as multi-copy sequences greater than 1 kb in size with more than 90% identity (Methods)—constituted 22.8–55.9% of the length of non-human ape Y chromosomes and only 4.0–7.2% of the X chromosomes (Fig. 1c and Supplementary Table 17). Segmental duplication coverage was almost two times higher on the Y chromosomes of *Pan* and *Pongo* lineages than of the other ape lineages (average 48.7% versus 26.6%, $P = 0.057$, Mann–Whitney $U$ test). We found

little evidence of lineage-specific segmental duplications on the X chromosome, but observed a gain of up to 2.2 Mb of interchromosomal segmental duplications in the T2T assembly compared with previous X assemblies[16–18]. Segmental duplications largely overlapped ampliconic regions and palindromes (Supplementary Note 5).

## Composition and methylation of repeats

Our comprehensive annotations (see Methods) revealed that 71–85% and 62–66% of Y and X chromosome lengths, respectively, consisted of repetitive elements (Fig. 4a and Supplementary Table 18)—comprising transposable elements, satellites and simple or low-complexity regions—compared with only 53% of the human T2T autosomal length[27]. On the Y chromosome, the repetitive element content (Fig. 4a and Supplementary Tables 18 and 19), comprised mainly of satellites and simple or low-complexity regions, and distributions (Extended Data Fig. 2) varied greatly among species, substantially contributing to the length variation. The transposable element content was significantly higher in Y ancestral than Y ampliconic regions (approximately 65.6% versus 46.9%; $P < 0.001$, Mann–Whitney $U$ test; Supplementary Fig. 8 and Supplementary Table 20), reflecting the absence of recombination in the Y ancestral regions and frequent intrachromosomal recombination in the Y ampliconic regions[5,9]. On the X chromosome, the transposable element content (Fig. 4a and Supplementary Table 18), comprising mainly retroelements and enriched for long interspersed elements[28] (Supplementary Table 19), and distributions (Extended Data Fig. 2) were similar among species. Notable exceptions included the expansion of alpha satellites at the non-centromeric regions in siamang[29],

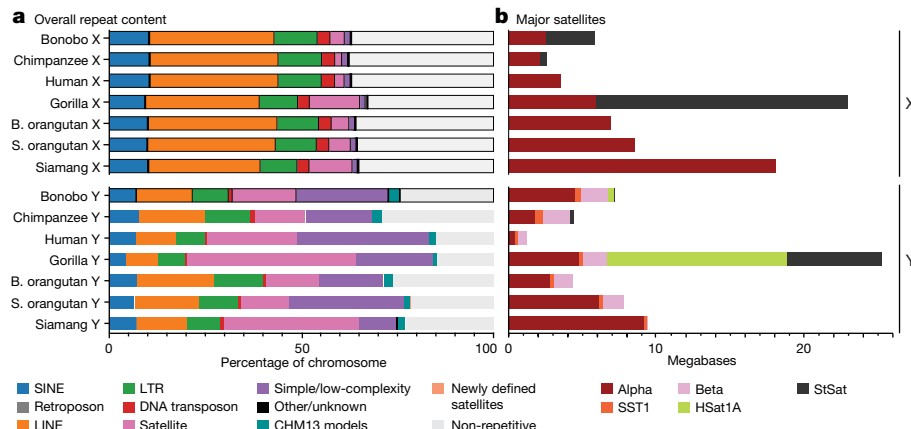

**Fig. 4 | Repeats on ape sex chromosomes. a**, Repeat annotations across each ape sex chromosome are depicted as a percentage of total nucleotides. Previously uncharacterized human repeats derived from the CHM13 genome analyses are shown in teal. Newly defined satellites (Methods) are depicted in light orange. **b**, The amount of DNA on each sex chromosome comprising canonical satellites, with each satellite represented by a different colour. LINE, long interspersed nuclear element; LTR, long terminal repeat; SINE, short interspersed nuclear element.

of the HSat1A satellite (also known as SAR) in non-human African apes, and of subtelomeric arrays of the StSat satellite (also known as pCht) in gorilla[30] (Fig. 4b and Extended Data Fig. 2). The transposable element content of X ancestral regions was significantly lower than that of Y ancestral regions (approximately 59.3% versus 65.6%; $P < 0.001$, Mann–Whitney $U$ test; Supplementary Fig. 8 and Supplementary Table 20) and significantly higher than that of Y ampliconic regions (approximately 46.9%; $P < 0.001$, Mann–Whitney $U$ test), consistent with different recombination rates among these regions. PARs maintained a similar repeat content and distribution across apes (Extended Data Fig. 2, Supplementary Fig. 8 and Supplementary Table 20).

We identified previously unknown composite repeats (a total of 13; Supplementary Fig. 9 and Supplementary Tables 21 and 22), variants of *DXZ4* repeats (a total of 2) and satellites (a total of 33; Supplementary Fig. 10 and Supplementary Table 23). The previously unknown satellites accounted for an average of 317 kb and 61 kb on each X and Y chromosome, respectively. Variable transposable element types and satellite arrays, including previously unknown satellites, expanded in a lineage-specific manner (Fig. 4a,b, Extended Data Fig. 2, Supplementary Fig. 11 and Supplementary Tables 24 and 25) either via intrinsic transposable element mobility or through other mechanisms. For example, the bonobo-specific satellite Ariel flanked PAR2 in a 318-unit array on the X chromosome and a 134-unit array on the Y chromosome (Supplementary Note 3). Lineage-specific expansions on the Y chromosome contributed more to interspecies variation than those on the X chromosome, but had similar patterns for both sex chromosomes between closely related species (Supplementary Note 6).

Our T2T assemblies enabled us to explore the distribution of motifs able to form non-B-DNA structures—A-phased repeats, direct repeats, G-quadruplexes, inverted repeats, mirror repeats, short tandem repeats and Z-DNA[31]—which have been implicated in numerous cellular processes, including replication and transcription[32]. Such motifs (see Methods) covered 6.3–8.7% of the X chromosome and 10–24% of the Y chromosome (Supplementary Table 26, Supplementary Fig. 12 and Methods). Each non-B-DNA motif type usually occupied a similar fraction and was located in similar regions of the X chromosomes among species, with direct repeats frequently located at the subtelomeric regions and inverted repeats at the centromeric regions. By contrast, the Y chromosomes exhibited a wide range of variation in content and location of different non-B-DNA types. Non-B-DNA was frequently enriched at satellites (Supplementary Fig. 13 and Supplementary Table 27), suggesting functional roles. For instance, the LSAU satellite[33] exhibited overrepresentation of G-quadruplexes, where they might function as mediators of epigenetic modifications[34],

consistent with variable methylation levels at this satellite among apes[35]. We also observed enrichment of inverted repeats at alpha satellites, consistent with the suggested role of non-B-DNA in centromere formation[36].

Given the strong effects of DNA methylation on repetitive elements and genome composition, we analysed 5-methylcytosine DNA methylation (hereafter referred to as methylation) patterns across ape sex chromosomes using long-read data mapped to these T2T assemblies. Previous studies suggested that in females, the inactive X chromosome may have lower global methylation than the active X chromosome[37,38], which is transcriptionally more active and less heterochromatic. We thus hypothesized that, in males, the Y chromosome, given its relative transcriptional inactivity[39] and high heterochromatin content, may have lower global methylation than the active X chromosome. In line with this expectation, the Y chromosome (excluding PARs) exhibited lower methylation levels than the X chromosome in long-range windows (Extended Data Fig. 3a and Supplementary Table 28). DNA methylation was higher for PAR1 than the rest of the X chromosome in all species (Extended Data Fig. 3a; Wilcoxon rank-sum test, $P$ values in Supplementary Table 28), which may be due to differences in recombination levels, as methylation is known to be increased in regions with high recombination rates[40]. Methylation differences between each PAR2 and the rest of the X chromosome were not significant (Supplementary Fig. 14a). Methylation levels were significantly higher in ampliconic regions, which undergo intrachromosomal recombination, than ancestral regions in chimpanzee, human and B. orangutan X chromosomes (Extended Data Fig. 3 and Supplementary Table 28), but were not significantly different between these two regions on the X chromosome of other species, and were lower in ampliconic than ancestral regions on the Y chromosome (Extended Data Fig. 3). Thus, the relationship between methylation and recombination might be different for intrachromosomal recombination versus interchromosomal recombination. Most groups of repetitive elements followed the general pattern of highest methylation in PAR1, intermediate in non-PAR X chromosome, and lowest in non-PAR Y chromosome (Extended Data Fig. 3b and Supplementary Table 28). The same pattern was observed in satellites (with the exception of human, which showed non-significant trends), despite their recent and frequent lineage-specific expansions. These patterns suggest rapid evolution of methylation on ape sex chromosomes.

## Evolution of centromere and rDNA arrays

We next examined the evolution of centromeres on X (cenX) and Y (cenY) chromosomes. Previous studies indicated that primate

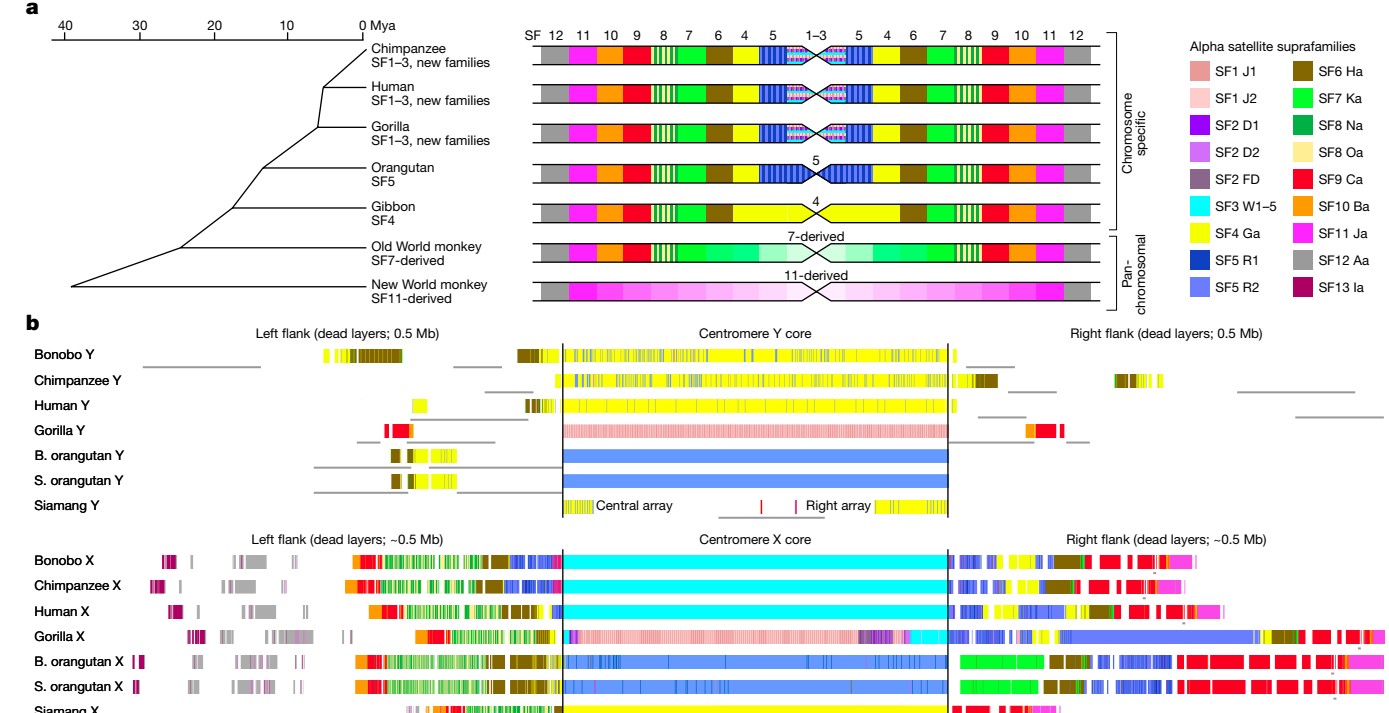

**Fig. 5 | Centromeres on ape sex chromosomes. a**, Left, active alpha satellite suprafamilies (SFs) on the primate phylogenetic tree. Active centromeres in each chromosome have different higher-order repeats in chromosome-specific organization and similar repeats in pan-chromosomal organization. Right, centromeres for each branch (not to scale) with alpha satellite suprachromosomal family composition of the active core indicated in the middle and of the dead flanking layers on the sides. Each branch has one or more alpha satellite suprachromosomal family fewer than in African apes but may also have layers not shared with human (indicated by hues of the same colour). The African ape centromere cores are shown as horizontal bars of SF1–SF3 as each chromosome usually has one alpha satellite suprafamily, which differs with each chromosome. **b**, The UCSC Genome Browser tracks of alpha satellite suprafamily composition of centromere cores and flanks for cenY and cenX (not to scale). CenX is surrounded by stable vestigial layers (that is, the remnants of ancestral centromeres), whereas cenY has a 'naked' centromere devoid of such layers. Thin grey lines under the tracks show overlaps with segmental duplications. In gorilla cenX, SF3 was replaced by SF2 and then by SF1 (see details in Supplementary Note 7).

centromere sequences underwent repeated remodelling cycles, in which new variants of 171-bp alpha satellite repeat monomers emerged and expanded within progenitor arrays, whereas vestigial layers of old displaced centromeres in the flanks degraded and shrank[41,42] (Fig. 5a). Indeed, each major primate lineage has active centromeres corresponding to a different alpha satellite suprachromosomal family (SF) group. Accordingly, cenXs in African apes are composed of 'younger' SF1–3 (Fig. 5b), whereas the 'older' SF5 and yet older SF4 form active centromeres in *Pongo* and siamang, respectively. Further, active arrays on cenX were flanked by older SF vestigial layers in all apes studied[43,44] (for example, by SF5, SF4 and SF6–11 in African apes; Fig. 5b). In contrast to cenX, whose chromosomal position has been stable throughout primate evolution, the chromosomal position of cenY is variable and lacks older flanking layers (Fig. 5b). CenY is defined by an older SF4 in human and *Pan*[8,45], rather than the younger SF1–3 typical of cenX and other African ape centromeres. This 'lagging' pattern was not observed in other ape cenYs, which aligned with expectations (Fig. 5b). For example, cenY in gorilla is defined by SF1, and as is typical of the younger SF1–3, contains CENP-B boxes (Supplementary Fig. 15a,e)—motifs that are important for the binding of centromere protein B, a key component of the inner kinetochore[46]. CENP-B boxes are absent in the SF4 arrays in human and *Pan* cenY, which can affect centromere function[46].

Ape centromeres consist of higher-order repeats (HORs), in which subsets of ordered alpha satellite monomers are arranged as a larger repeating unit with high sequence similarity between copies (Supplementary Tables 29 and 30, Supplementary Note 7 and Methods). HORs on cenX and cenY are lineage-specific in apes, with the exception of the shared cenX HOR in human and *Pan*. In closely related species (chimpanzee and bonobo, or the two orangutan species) we observed the same HORs; however, their arrays differed in length, structural variant composition and centromere dip regions, the signature methylation pattern that marks the kinetochore location[44,47] (Extended Data Fig. 4 and Supplementary Fig. 15b,c). Further classification of HORs revealed species-specific HOR haplotypes[43,44] with subtle signatures of array remodelling, comparable to the turnover of alpha satellite suprachromosomal families (Extended Data Fig. 4, Supplementary Fig. 15d and Supplementary Note 7). Finally, SF4 alpha satellite arrays were identified in the siamang in both centromeres and subtelomeric regions[29]. In contrast to the highly similar subtelomeric arrays (Supplementary Fig. 15f), the non-telomeric arrays in siamang were chromosome-specific, similar to these in other apes[29,42].

rDNA arrays were found on the Y chromosomes of siamang, S. orangutan and B. orangutan[48,49], but not on any X chromosomes (Fig. 1c). Individual UL-ONT reads confirmed the presence of three copies for S. orangutan and one copy for B. orangutan, but were not long enough to span the siamang array. Instead, fluorescent in situ hybridization (FISH) was used to estimate the size of the siamang array at 16 copies and to confirm the absence of rDNA signal on all other sex chromosomes (Extended Data Fig. 5a,b, Supplementary Fig. 16, Supplementary Table 31 and Methods). Evidence of active 45S transcription was found for both the siamang and S. orangutan arrays, whereas the single B. orangutan unit appeared silent (Extended Data Fig. 5c–e). Beyond the genomes assembled here, we also found rDNA on the Y chromosomes of white-cheeked and black crested gibbons (Supplementary Note 8).

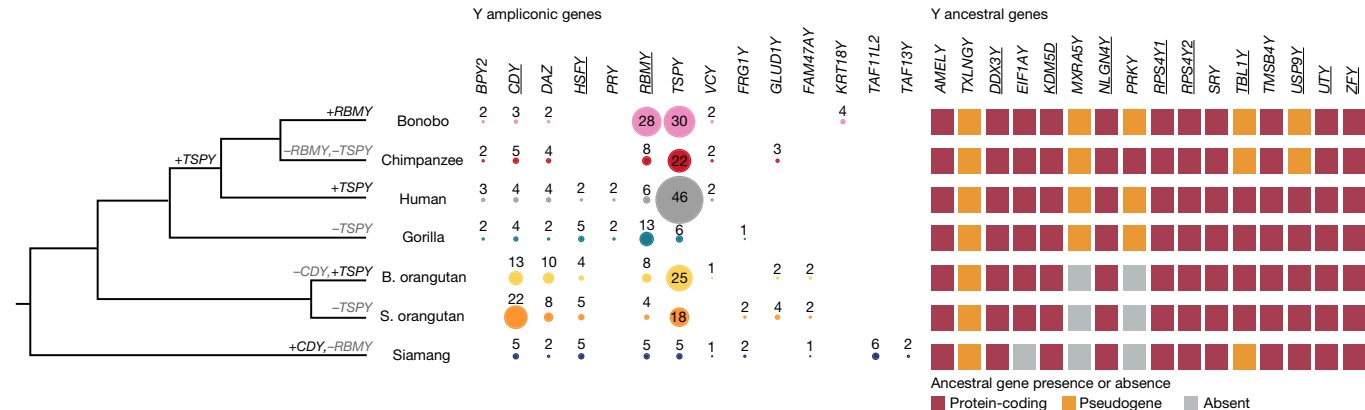

**Fig. 6 | Gene evolution on the Y chromosome.** Significant gains and losses in ampliconic gene copy number (Supplementary Note 10) are shown on the phylogenetic tree. Copy numbers of ampliconic genes are indicated with numbers and by circle size; no circle indicates absence of annotated protein-coding copies. Presence, pseudogenization or absence (that is, deletion) of ancestral (X-degenerate) genes are shown by squares of different colours.

Genes showing signatures of purifying selection (Methods) are underlined. *XKRY* was found to be a pseudogene in all species studied and is therefore not shown. The protein-coding status of *PRY* was confirmed for human[8], and we found evidence of expression of a similar transcript in gorilla (Supplementary Table 36b). The *RBMY* gene family harboured two distinct gene variants, each present in multiple copies in *Pongo* (Supplementary Fig. 19).

## Protein-coding genes

Our gene annotations (Supplementary Table 32 and Methods) indicated the presence of a high percentage of BUSCO genes on the X chromosomes (Supplementary Table 33), and of most previously known Y chromosome genes (Fig. 6). We manually curated Y chromosome genes (Methods) and validated the copy number of several multi-copy gene families on the Y chromosome with droplet digital PCR (ddPCR; Supplementary Tables 34 and 35). As a rule, genes were single-copy in ancestral regions and multi-copy in ampliconic regions (Supplementary Tables 36 and 37). On the X chromosome, gene density was around 2.5- to 5-fold higher in the ampliconic regions than in ancestral regions (16–25 versus 5.3–6.1 genes per Mb; Fig. 3b and Supplementary Table 38) and was higher still in palindromes (27–34 genes per Mb; Fig. 3b). Palindromes shared among species contained many housekeeping gene families (for example, *CENPVL*, *H2A* and *FAM156*; Supplementary Tables 37 and 38). Gene density was uniformly lower on the Y chromosome than on the X chromosome (Fig. 3b), with a low density in both ancestral (2.0–4.5 genes per Mb) and ampliconic (2.7–5.7 genes per Mb) regions.

The ancestral (or 'X-degenerate') gene content on the Y chromosome was generally well conserved (Fig. 6 and Supplementary Note 9), with the exception of *TXLNGY*, *MXRA5Y* and *PRKY*, which were pseudogenized or lost in all or nearly all studied apes (Supplementary Table 39). Ten ancestral genes were present in all studied apes, and 9 out of 13 ancestral genes analysed exhibited a signature of purifying selection ($P \le 0.05$, likelihood ratio test (LRT); Supplementary Table 40)—that is, the nonsynonymous-to-synonymous rate ratio, $d_N/d_S$, was below 1 ($P \le 0.05$, LRT; Supplementary Table 40). Notably, all four ancestral genes found to be retained in eutherian mammals in another study[4] were present in apes, and three of them (*DDX3Y*, *UTY* and *ZFY*, but not *SRY*) had a $d_N/d_S$ of less than one.

Among multi-copy genes on the Y and X chromosomes, we detected ampliconic gene families, defined as families with at least two copies having ≥97% sequence identity at the protein level in at least one species (Supplementary Tables 36 and 37). Many of them were located in palindromes. The proportion of ampliconic among multi-copy gene families was lower on the X chromosome than on the Y chromosome (55 out of 123 versus 14 out of 20; $P = 0.0358$, chi-squared test). Nevertheless, we still found several copious ampliconic gene families on the X chromosome—*GAGE*, *MAGE* and *SPANX*—the products of which are expressed in testis (Supplementary Table 37).

Among Y ampliconic gene families, ten have been described previously[6,14] (*BPY2*, *CDY*, *DAZ*, *HSFY*, *PRY*, *RBMY*, *TSPY*, *VCY*, *FRG1* and *GLUD1*), with the majority functioning in spermatogenesis[6], and four (*FAM47AY*, *KRT18Y*, *TAF13Y* and *TAF11L2Y*) are described here (Fig. 6 and Supplementary Table 36). Some ampliconic gene copies were located at multiple palindromes and/or outside of palindromes (Extended Data Fig. 6 and Supplementary Table 41). We found episodes of significant lineage-specific expansions and contractions in the previously described ampliconic gene families (Fig. 6 and Supplementary Note 10); for example, *RMBY* expanded in bonobo, *CDY* expanded in S. orangutan, and *TSPY* expanded in human. These results for one individual per species are largely consistent with prior ddPCR results for multiple individuals per species[39]. *TSPY*—the only ampliconic gene family located in tandem arrays outside of palindromes in all species except bonobo and siamang (Supplementary Table 41)—had a high copy number in all species except gorilla and siamang (Fig. 6). A phylogenetic analysis identified mainly species-specific and genus-specific clades (Extended Data Fig. 7) with short branches for individual *TSPY* protein-coding copies, suggesting sequence homogenization due to recombination between palindrome arms and/or direct repeats[50]. The newly described ampliconic gene families had more limited species distribution and were usually less copious than the previously described families (Fig. 6). We found no evidence of positive selection acting on Y ampliconic gene families (Supplementary Table 40). A significant signal of purifying selection was detected for only three (*CDY*, *HSFY* and *RBMY*) out of seven gene families analysed ($P \le 0.05$, LRT; Supplementary Table 40). Congruous with an observation for human and macaque[5], apes had a lower group-mean $d_N/d_S$ for Y chromosome ancestral than for Y chromosome ampliconic genes (0.38 versus 0.69; joint model fit, LRT $P$ value < $10^{-10}$), suggesting stronger purifying selection acting on the Y chromosome ancestral genes.

The characteristic DNA methylation levels near the transcription start sites of protein-coding genes (Supplementary Fig. 14b,c) and their relationship with gene expression (Supplementary Fig. 14d) implies the importance of promoter hypomethylation in the regulation of gene expression[51] on both sex chromosomes. Because de novo genes—lineage-specific genes arising from non-coding sequences—have a role in fertility and frequently have testis-specific expression[52], they might emerge on the Y chromosome. Using our T2T assemblies, we indeed traced the emergence of two candidate Y-specific de novo genes—one in bonobo and one in siamang (Supplementary Note 11).

## Intraspecific ape diversity and selection

Our T2T assemblies enabled us to perform sex chromosome-wide analyses of great ape intraspecific diversity. Aligning short sequencing reads from 129 individuals across 11 subspecies (Supplementary Table 42a) to T2T and previous assemblies (see Methods), we detected a higher proportion of reads mapping and a lower mismatch rate to the T2T assemblies in most cases (Extended Data Fig. 8a, Supplementary Fig. 17a and Supplementary Table 42). The variants identified relative to the T2T assemblies contained fewer single nucleotide variants and small insertion–deletion homozygous variants (Supplementary Fig. 17b and Supplementary Table 42), which can arise from structural errors in the reference genome[53], and largely restored the expected site frequency spectrum (Extended Data Fig. 8b). However, eastern lowland and mountain gorillas still contained a substantial number of homozygous variants (Supplementary Fig. 17c), highlighting the need for additional species- and subspecies-specific references. Within the chimpanzee Y chromosome, the T2T assembly identified a more uniform read distribution and more variants due to the increased length (Extended Data Fig. 8c), as well as a 33-fold reduction in variants over an ampliconic region segment (Extended Data Fig. 8d), probably due to a collapse of this segment in the previous assembly.

Leveraging the more accurate and complete variant calls, we next studied the nucleotide diversity of the different species. Across the X chromosome, the diversity was higher for S. orangutans than for B. orangutans ($P < 0.001$, Mann–Whitney $U$ test; Extended Data Fig. 8e), in agreement with prior work[54]. In the *Pan* lineage, central chimpanzees retained the highest diversity ($P$ values ≤ 0.01, Mann–Whitney $U$ test). Nigeria–Cameroon and western chimpanzees had a relatively low diversity, probably signalling historical population bottlenecks[55]. The western lowland gorillas retained a higher diversity than the eastern lowland and mountain gorillas ($P$ values < 0.002, Mann–Whitney $U$ test), both of which have undergone a prolonged population decline[56]. In most subspecies studied, the Y chromosome exhibited a significantly lower diversity than the X chromosome ($P$ values ≤ 0.01, Mann–Whitney $U$ test; Extended Data Fig. 8e), as was reported in humans[57]. Among the great apes, bonobos displayed the highest diversity on the Y chromosome.

Of particular interest was putative selection on the Y chromosome, which can evolve rapidly owing to different levels of sperm competition among species[6] (Supplementary Table 2). We analysed combined chimpanzee and gorilla samples for nucleotide diversity and Tajima's $D$ and derived expected values from neutral simulations (Supplementary Note 12). In gorillas, the observed Y/X diversity ratio was considerably lower than in simulations. In chimpanzees, this ratio aligned with neutrality only at very low male effective population sizes. Because male effective population size is high in chimpanzees[58], this suggests selection reduced diversity on the Y chromosome in both species, consistent with reports for humans[57]. Tajima's $D$ results suggested that purifying selection drives this reduction in diversity on the Y chromosome in both species (Supplementary Note 12). Additionally, we identified 45 genes in gorilla and 81 genes in chimpanzee that overlap with candidate regions of selection (Supplementary Note 12). Finally, incorporating diversity information, we found no evidence of positive selection on ancestral genes on the Y chromosome in chimpanzee and gorilla (Supplementary Note 13).

## Discussion

Our complete assemblies have revealed the evolution of great ape sex chromosomes in unprecedented detail. In contrast to the X chromosome, the Y chromosome has undergone rapid evolution in all ape species. It has accumulated repetitive elements and experienced elevated rates of nucleotide substitutions, intrachromosomal rearrangements and segmental duplications, probably owing to the loss of recombination over most of its length. It also has reduced global levels of DNA methylation, linked to the low expression levels of many of its genes[39]. Because of this degradation, the Y chromosome has been suggested to be on its way towards extinction in mammals[2]. Our study suggests that it is still present in apes in part because it contains several protein-coding genes that are evolving under purifying selection, similar to observations for rhesus macaque[59]. Future studies should investigate non-coding genes and regulatory elements on the Y chromosome, which may be essential for males and further contribute to selective pressure.

Palindromes are thought to be critical for counterbalancing the degradation of the Y chromosome by enabling intrachromosomal recombination and gene conversion[10]. Thus, we expected palindromes on the Y chromosome to be conserved, but instead found many of them to be lineage-specific. Rapid acquisition of new Y chromosome palindromes might be due to random genetic drift, which should be strong on the Y chromosome because of its small effective population size[60], and/or owing to species-specific selection. Our analysis of Y chromosome ampliconic genes, which are primarily located in palindromes and have a role in spermatogenesis, did not provide evidence of species-specific selection. Instead, we found a higher ratio of nonsynonymous-to-synonymous mutations for ampliconic versus single-copy genes, consistent with either relaxation of functional constraints or a higher rate of fixation of beneficial mutations due to gene conversion in ampliconic genes[5]—possibilities that should be distinguished by future analyses. Notably, copies of some Y ampliconic genes were present at multiple locations on the Y chromosome, and not just within a single palindrome or tandem repeat, providing an additional mechanism safeguarding genes on this non-recombining chromosome. The X chromosome also undergoes less recombination than the autosomes as, outside of PARs, it does not recombine in males. We found that it has utilized some of the same strategies to preserve its genetic content, including maintaining palindromes in all apes studied and having ampliconic gene copies at multiple locations.

In addition to gene amplifications, a variety of lineage-specific satellite expansions were observed in the apes, with some specific to the Y chromosome (for example, HSat1A in the gorilla Y chromosome) and some shared between X and Y chromosomes (for example, alpha satellite in siamang). These observations prompt a question about the functionality of these satellites, including those that are enriched in non-B-DNA, since such structures may serve as binding sites for protein regulators[32] and may be involved in defining centromeres[36]. Satellites on the *Drosophila* sex chromosomes contribute to regulation of gene expression of autosomal genes[61] and to reproductive isolation among species[62]; similar phenomena should be investigated in apes. Further work is needed to clarify the potential role of satellites in recombination. In some of the species studied here, subtelomeric satellites distal to the PAR were shared between X and Y chromosomes. If recombination occurs within these satellites, our current PAR annotation will need to be expanded to include them. Additionally, the putative PAR2 sequence discovered in bonobo is flanked by an Ariel satellite that may serve as a *cis*-acting factor for increased double-strand break formation, as was found for a mo-2 minisatellite in mouse[63]. However, the bonobo PAR2 sequence was also found at the ends of several autosomes (Supplementary Note 3) and thus might act as a general facilitator of recombination or represent a subtelomeric duplication[64]. The presence of active rDNA arrays on the Y chromosomes of some species also hints at ectopic recombination between the Y chromosome and the short arms of the rDNA-bearing acrocentric chromosomes[8,65].

Mapping short reads from multiple non-human ape individuals revealed intriguing patterns of diversity and highlighted the critical need for collecting additional samples. Further intraspecific studies, comparing the complete sex chromosomes of multiple individuals per species (as was recently done for the human Y chromosome[66]) and subspecies are required to reveal the full landscape of ape sex chromosome evolution. Such studies will be useful for investigating

sex-specific dispersal and will greatly inform conservation efforts in non-human ape species, all of which are endangered. In humans, both sex chromosomes are important for reproduction[1,2], genes on the X chromosome are also critical for cognition[2], abnormal X chromosome gene dosage underlies female bias in autoimmune disorders[67], and X-linked mutations are responsible for 10% of Mendelian disorders[68], even though the X chromosome constitutes only around 5% of the genome[20]. Thus, we expect these T2T assemblies to be pivotal for understanding disease-causing mutations and human-specific traits.

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

Kateryna D. Makova[1,27 ✉], Brandon D. Pickett[2,27], Robert S. Harris[1,27], Gabrielle A. Hartley[3,27], Monika Cechova[4,27], Karol Pal[1,27], Sergey Nurk[2], DongAhn Yoo[5], Qiuhui Li[6], Prajna Hebbar[4], Barbara C. McGrath[1], Francesca Antonacci[7], Margaux Aubel[8], Arjun Biddanda[6], Matthew Borchers[9], Erich Bornberg-Bauer[8,10], Gerard G. Bouffard[2], Shelise Y. Brooks[2], Lucia Carbone[11,12], Laura Carrel[13], Andrew Carroll[14], Pi-Chuan Chang[14], Chen-Shan Chin[15], Daniel E. Cook[14], Sarah J. C. Craig[1], Luciana de Gennaro[7], Mark Diekhans[4], Amalia Dutra[2], Gage H. Garcia[5], Patrick G. S. Grady[3], Richard E. Green[4], Diana Haddad[16], Pille Hallast[17], William T. Harvey[5], Glenn Hickey[4], David A. Hillis[18], Savannah J. Hoyt[3], Hyeonsoo Jeong[5], Kaivan Kamali[1], Sergei L. Kosakovsky Pond[19], Troy M. LaPolice[1], Charles Lee[17], Alexandra P. Lewis[5], Yong-Hwee E. Loh[18], Patrick Masterson[16], Kelly M. McGarvey[16], Rajiv C. McCoy[6], Paul Medvedev[1], Karen H. Miga[4], Katherine M. Munson[5], Evgenia Pak[2], Benedict Paten[4], Brendan J. Pinto[20], Tamara Potapova[9], Arang Rhie[2], Joana L. Rocha[21], Fedor Ryabov[22], Oliver A. Ryder[23], Samuel Sacco[4], Kishwar Shafin[14], Valery A. Shepelev[24], Viviane Slon[25], Steven J. Solar[2], Jessica M. Storer[3], Peter H. Sudmant[21], Sweetalana[1], Alex Sweeten[2,6], Michael G. Tassia[6], Françoise Thibaud-Nissen[16], Mario Ventura[7], Melissa A. Wilson[20], Alice C. Young[2], Huiqing Zeng[1], Xinru Zhang[1], Zachary A. Szpiech[1], Christian D. Huber[1], Jennifer L. Gerton[9], Soojin V. Yi[18], Michael C. Schatz[6], Ivan A. Alexandrov[25], Sergey Koren[2], Rachel J. O'Neill[3], Evan E. Eichler[5,26 ✉] & Adam M. Phillippy[2 ✉]

[1]Penn State University, University Park, PA, USA. [2]National Human Genome Research Institute, National Institutes of Health, Bethesda, MD, USA. [3]University of Connecticut, Storrs, CT, USA. [4]University of California Santa Cruz, Santa Cruz, CA, USA. [5]University of Washington School of Medicine, Seattle, WA, USA. [6]Johns Hopkins University, Baltimore, MD, USA. [7]Università degli Studi di Bari Aldo Moro, Bari, Italy. [8]University of Münster, Münster, Germany. [9]Stowers Institute, Kansas City, MO, USA. [10]MPI for Developmental Biology, Tübingen, Germany. [11]Oregon Health and Science University, Portland, OR, USA. [12]Oregon National Primate Research Center, Hillsboro, OR, USA. [13]Penn State University School of Medicine, Hershey, PA, USA. [14]Google, Mountain View, CA, USA. [15]Foundation of Biological Data Sciences, Belmont, CA, USA. [16]National Center for Biotechnology Information, National Library of Medicine, National Institutes of Health, Bethesda, MD, USA. [17]The Jackson Laboratory for Genomic Medicine, Farmington, CT, USA. [18]University of California Santa Barbara, Santa Barbara, CA, USA. [19]Temple University, Philadelphia, PA, USA. [20]Arizona State University, Tempe, AZ, USA. [21]University of California Berkeley, Berkeley, CA, USA. [22]Masters Program in National Research, University Higher School of Economics, Moscow, Russia. [23]San Diego Zoological Society, San Diego, CA, USA. [24]Institute of Molecular Genetics, Moscow, Russia. [25]Tel Aviv University, Tel Aviv, Israel. [26]Howard Hughes Medical Institute, University of Washington, Seattle, WA, USA. [27]These authors contributed equally: Kateryna D. Makova, Brandon D. Pickett, Robert S. Harris, Gabrielle A. Hartley, Monika Cechova, Karol Pal. ✉e-mail: kdm16@psu.edu; eee@gs.washington.edu; adam.phillippy@nih.gov

## Methods

### Sequencing and assemblies

**Sequencing.** We built a collection of male fibroblast and lymphoblastoid cell lines for these species (Supplementary Table 3 and Supplementary Notes 1 and 2), each karyotyped (Supplementary Fig. 1) to confirm absence of large-scale chromosomal rearrangements, and isolated high-molecular-weight DNA from them. Whole-genome DNA sequencing was performed using three different sequencing technologies. To obtain long and accurate reads, Pacific Biosciences (PacBio) HiFi sequencing was performed on Sequel II with a depth of >60×. To obtain ultra-long (>100-kb) reads, Oxford Nanopore Technologies (ONT) sequencing was performed on PromethION to achieve ≥100 Gb (≥29× depth). To assist with assemblies, paired-end short-read sequencing was performed on Hi-C (Dovetail Omni-C from Cantata Bio) libraries sequenced on Illumina NovaSeq 6000, targeting 400 M pairs of 150-bp reads (≥30× depth) per sample. For bonobo and gorilla parents, we generated paired-end short reads on an Illumina NovaSeq 6000 to achieve ≥518 million pairs of 151-bp reads (≥51× depth) for each sample. Full-length transcriptome sequencing was performed on testes tissue from specimens other than the T2T genome targets (Supplementary Table 43) using PacBio Iso-Seq on up to three SMRT (8 million) cells using Sequel II.

**Assemblies.** The complete, haplotype-resolved assemblies of chromosomes X and Y were generated using a combination of Verkko[21] and expert manual curation. Haplotype-specific nodes in the Verkko graphs were labelled using parental-specific *k*-mers when trios were available (bonobo and gorilla) or Hi-C binned assemblies in the absence of trios (chimpanzee, orangutans and siamang). Haplotype-consistent contigs and scaffolds were automatically extracted from the labelled Verkko graph, with unresolved gap sizes estimated directly from the graph structure (further details in ref. 21).

During curation, the primary component(s) of chromosomes X and Y were identified on the basis of the graph topology as visualized in Bandage[71] and using MashMap[72] alignments of the assembly to the CHM13 human reference[20]. Several X and Y chromosomes were automatically completed by Verkko and required no manual intervention; for the remainder, manual interventions were used (Supplementary Table 6). Using available information such as parent-specific *k*-mer counts, depth of coverage, and node lengths, some artifactual edges could be removed and simple non-linear structures resolved. For more complex cases, ONT reads aligned through the graph were used to generate multiple candidate resolutions, which were individually validated to select the one with the best mapping support. Disconnected nodes due to HiFi coverage gaps were joined and gap-filled using localized, ONT-based Flye[73] assemblies. The resulting gapless, telomere-to-telomere (T2T) assemblies were oriented based on MashMap alignments to the existing reference genomes of the same or related species (Supplementary Table 7); in v1.1 of the assemblies, all chromosomes were oriented to start with PAR1.

To validate the T2T assemblies of chromosomes X and Y, we aligned all available read data (Supplementary Table 4) to the assemblies to measure agreement between the assemblies and raw sequencing data. Specific alignment methods differed for the various data types (Supplementary Methods), but the general principles from McCartney et al.[74] were followed. Validation of the assemblies was done in multiple ways to assess assembly completeness and correctness. Coverage analysis, erroneous *k*-mers, and haplotype-specific *k*-mers (for the two trios) were manually inspected using Integrated Genome Viewer[75] (IGV), and assembly quality verification was calculated using Merqury[76]. The completeness of each chromosome was confirmed by the identification of telomeric arrays on each end and uniform coverage of long-read mappings, with an absence of clipped reads or other observable mapping artifacts.

### Alignments

**Pairwise alignments.** To compute the percentage of sequences aligned and to study structural variants and segmental duplications, the pairwise alignment of the human chromosome X and Y was performed against each of chromosome X and Y of the six ape species using minimap2.24[77]. To support other analyses, lastz[78] was used to compute pairwise alignments of X and Y chromosomes for each species.

**Multi-species whole-chromosome alignments.** To estimate the substitution rates on the X and Y chromosomes, we used CACTUS[69] to generate multiple alignments for the seven species, first for the X sequences, and separately for the Y sequences. Sequences were soft-masked using repeat annotations (see section on Satellite and repeat analysis below). We provided CACTUS with a guide tree, (((((bonobo,chimp),human),gorilla),(sorang,borang)),gibbon), but did not provide branch lengths.

### Nucleotide substitution analysis

**Nucleotide substitution frequency analysis.** Substitution rates were estimated (separately for the X and the Y chromosomes) for CACTUS alignment blocks containing all seven species with the REV model implemented in PHYLOFIT[79].

**Nucleotide substitution spectrum analysis.** Substitution spectrum analysis was conducted using 13-way CACTUS[69] alignments, which, in addition to the 7 studied species, include 6 ancestral species sequences reconstructed by CACTUS[69]. Triple-nucleotide sequences with 5′ base identical among 13 sequences and 3′ base identical among 13 sequences were used for downstream substitution spectrum analysis. For each branch, 96 types of substitution (depending on tri-nucleotide context) were grouped into 6 types based on the middle base substitutions (C>A, C>G, C>T, T>A, T>C and T>G). To compare the distribution of substitution types between chromosome X and chromosome Y, we applied *t*-tests to the proportions of each substitution type per branch, using Bonferroni correction for multiple testing.

### Duplications and structural variants

**Segmental duplications.** The segmental duplication content in humans and non-human primates was identified using SEDEF (v1.1)[80] based on the analysis of genome assemblies soft-masked with TRF v.4.0.9[81], RepeatMasker[82], and Windowmasker (v2.2.22)[83]. The segmental duplication calls were additionally filtered to keep those with sequence identity >90%, length >1 kb, and satellite content <70%. Lineage-specific segmental duplications were defined by comparing the putative homologous segmental duplication loci, defined as containing 10-kb syntenic sequence flanking the segmental duplication. The lineage-specific segmental duplications of each species were identified on the basis of non-orthologous locations in the genomes.

**Structural variants.** Structural variants were identified against the human reference genome CHM13v2.0 via minimap (v2.24) pairwise alignment of ape chromosomes against the human chromosome X and Y[77,84]; 50-bp to 300-kb sized structural variants with PAV[85]. Larger events were identified and visually inspected using the Saffire structural variant calling pipeline (https://github.com/wharvey31/saffire_sv). The human-specific structural variants were identified by intersecting the variant loci of six ape species; deletions in the six ape species relative to human reference chromosome as putative human-specific insertions, and insertions as putative human-specific deletions. The phylogenetic branch of origin of each structural variant was predicted using maximum parsimony. As a limitation of this analysis, the structural variants for branches including ancestors of the reference species (human ancestors—that is, human–chimpanzee–bonobo, human–chimpanzee–bonobo–gorilla and

human–chimpanzee–bonobo–gorilla–orangutan common ancestors) were not computed.

## Palindromes and ampliconic regions

**Palindrome detection and grouping.** We developed palindrover to screen the X and Y chromosomes for palindromes with ≥98% sequence identity, length ≥8 kb and spacer ≤500 kb, only keeping candidates with <80% of repetitive content. After aligning the arms with lastz[78] (alignments with identity <85%, gaps >5%, <500 matched bases, or covering less than 40% of either arm, were discarded), we identified orthologous palindromes and grouped paralogous palindromes on the same chromosome. Grouping palindromes into clusters was done via transitive closure of aligning (sequence sharing) palindrome pairs—if palindrome pair A and B and pair B and C were identified, all palindromes A, B, and C were considered to be in one cluster.

**Overview of the workflow for sequence class annotations.** We annotated sequence classes following[6], with modifications. First, PARs and satellite repeat tracks were created (by aligning X and Y chromosomes for PARs, and by merging adjacent (within 1 kb) RepeatMasker[82] annotation spanning >0.25 Mb). Next, ampliconic regions were identified as a union of palindromes and regions with high intrachromosomal similarity (that is, similar to other locations within non-PAR, here identified as consecutive 5-kb windows mapping with ≥50% identity to the repeat-masked chromosomes using blastn from BLAST+ v.2.5.0[86,87], excluding self-alignments, and spanning >90 kb). The remaining subregions on the Y were annotated as ancestral or ampliconic if overlapping respective genes. Subregions nested within two matching classes were annotated as such.

## Satellite and repeat analysis

**Satellite and repeat annotations.** We produced comprehensive repeat annotations for both X and Y chromosomes across the ape lineage by integrating a combination of known repeats and models identified in human CHM13[20,27] and T2T-Y[8], and de novo repeat curation (Supplementary Table 18). To identify canonical and novel repeats on chromosomes X and Y, we utilized the previously described pipeline[27], with modifications to include both the Dfam 3.6[88] and Repbase (v20181026)[89] libraries for each species during RepeatMasker[90] annotation. A subsequent RepeatMasker run was completed to include repeat models first identified in the analysis of T2T-CHM13 (Supplementary Table 44), and the resulting annotations were merged. To identify and curate previously undefined satellites, we utilized additional TRF[81] and ULTRA[91] screening of annotation gaps >5 kb in length. To identify potential redundancy, satellite consensus sequences generated from gaps identified in each species were used as a RepeatMasker library to search for overlap in the other five analysed primate species. Consensus sequences were considered redundant if there was a significant annotation overlap in the RepeatMasker output. Subsequently, final repeat annotations were produced by combining newly defined satellites and 17 variants of pCht/StSat derived from Cechova et al.[92] and merging resulting annotations. Newly defined satellites that could not be searched using RepeatMasker[90] due to complex variation were annotated using TRF[81] and manually added. Tandem composite repeats were identified using self-alignment dot plots and subsequently curated using BLAT[93] to identify unit lengths and polished using a strategy defined in ref. 94. Composite repeats were compiled in a distinct repeat annotation track from canonical repeat annotations.

Lineage-specific insertions or expansions were characterized by identifying unaligned regions from CACTUS alignments of the seven primate X and Y chromosomes with halAlignExtract[95]. Unaligned regions were filtered by length and for tandem repeats using TRF[81] and ULTRA[91]. RepeatMasker[90] was used to identify the content of the lineage-specific insertions/expansions using the approach described above.

**Non-B-DNA annotations.** G-quadruplex motifs were annotated with Quadron[96], and other types of non-B-DNA motifs were annotated with gfa (https://github.com/abcsFrederick/non-B_gfa). To compute non-B-DNA density, we used the coverage command in bedtools to count the number of overlaps between each 100-kb window and non-B-DNA motifs. We used the glm function implemented in R to perform simple and multiple logistic regression to evaluate the relationship between non-B-DNA density and sequences gained by the new assemblies. The non-B-DNA enrichment analysis for satellites is described in Supplementary Methods.

**Centromere analysis.** To analyse centromeres, we annotated alpha satellites and built several tracks at the UCSC Genome Browser (https://genome.ucsc.edu/s/fedorrik/primatesX and https://genome.ucsc.edu/s/fedorrik/primatesY): (1) Suprachromosomal Family tracks using human-based annotation tools[44] and utilizing score/length thresholds of 0.7, 0.3, and no threshold; (2) alpha satellite-strand track; (3) HOR track using species-specific tools specifically designed for this project (https://github.com/fedorrik/apeXY_hmm) and methods described in ref. 44; (4) structural variation (that is, altered monomer order) tracks in HORs; (5) CENP-B sites visualized by running a short match search with the sequence YTTCGTTGGAARCGGGA. Other methods are described in Supplementary Methods and Supplementary Note 7.

## Gene annotations and analysis

**Gene annotations at the NCBI.** The de novo gene annotations of the 6 primate assemblies were performed by the NCBI Eukaryotic Genome Annotation Pipeline as previously described for other genomes[97,98], between 20 March and 31 May 2023. The annotation of protein-coding and long non-coding genes was derived from the alignments of primate transcripts and proteins queried from GenBank and RefSeq, and same-species (but usually not the same-individual) RNA-sequencing (RNA-seq) reads and PacBio Iso-Seq queried from the Sequence Read Archive to the WindowMasker[83] masked genome. cDNAs were aligned to the genomes using Splign[99], and proteins were aligned using ProSplign. The RNA-seq reads (Supplementary Data 4), ranging from 673 million (*P. pygmaeus*) to 7.3 billion (*P. troglodytes*) were aligned to the assembly using STAR[100], while the Iso-seq reads (ranging from none for *S. syndactylus* to 27 million for *G. gorilla*) were aligned using minimap2[77]. Short non-coding RNAs, rRNAs, and tRNAs were derived from RFAM[101] models searched with Infernal cmsearch[102] and tRNAscan-SE[103], respectively.

**Gene annotations at the UCSC.** Genome annotation was performed using the Comparative Annotation Toolkit (CAT)[104]. First, whole-genome alignments between the primate (gorilla, chimpanzee, bonobo, S. orangutan, B. orangutan and siamang) and human GRCh38, and T2T-CHM13v2 genomes were generated using CACTUS[69], as described above. CAT then used the whole-genome alignments to project the UCSC GENCODEv35 CAT/Liftoff v2 (https://cgl.gi.ucsc.edu/data/T2T-primates-chrXY/chm13.draft_v2.0.gene_annotation.gff3) annotation set from CHM13v2 to the primates. In addition, CAT was given Iso-seq FLNC data to provide extrinsic hints to the Augustus PB (PacBio) module of CAT, which performs ab initio prediction of coding isoforms. CAT was also run with the Augustus Comparative Gene Prediction (CGP) module, which leverages whole-genome alignments to predict coding loci across many genomes simultaneously (that is, gene prediction). CAT then combined these ab initio prediction sets with the human gene projections to produce the final gene sets and UCSC assembly hubs used in this project.

**Curation and analysis of ancestral genes.** For the Y chromosome, we collected annotations from the NCBI Eukaryotic Genome Annotation Pipeline (RefSeq), CAT and Liftoff. We extracted ancestral gene annotations from each and mapped them onto the Y chromosome sequence for each in Geneious[105]. We identified that every gene was present and

manually curated an annotation set with the most complete exonic complement across annotations. We extracted all CDS regions for each gene and aligned them. For the X chromosome, we extracted ancestral gene copies from the RefSeq annotations using gffread[106] and aligned them. All alignments were examined and curated by eye, and missing genes and exons were confirmed using BLAST[87]. All present genes were aligned to their orthologues and their gametologues, where we identified genes with significant deviations (truncations of 20% or greater) relative to known (functional) Y copies in other ape species, or their X chromosome counterpart, as pseudogenes (Supplementary Table 39). These alignments were also used to identify gene conversion events using GeneConv[107] and to detect selection (see section Gene-level selection using interspecific fixed differences below).

**Detection of multi-copy and ampliconic gene families.** We used blastp for all protein sequences of all protein-coding genes (as annotated by NCBI) against a blast database built from these sequences, separately for the X and the Y chromosome. To infer homology we used a cutoff of 50% sequence identity of at least 35% of protein lengths[108]. We then clustered genes into multi-copy families using a simplified single linkage approach (if genes A and B shared sequence identity and so did genes B and C, we created a group of genes A, B and C). To overcome the shortcomings of this method, we removed gene clusters where no genes within one species shared high enough sequence identity.

For each multi-copy gene family we collected the counts of occurrences of gene copies, the sequence classes assigned to the regions where these copies occur, and all pairwise identities of gene copies within one species (Supplementary Tables 36 and 37). Among multi-copy gene families we then delineated ampliconic families as those that had ≥97% protein sequence identity between at least two copies in a family in at least one species, which we chose because it was a natural breakpoint in the pairwise sequence identity distribution for Y multi-copy genes (Supplementary Fig. 20). This method identified all previously known Y ampliconic gene families (*BPY2*, *CDY*, *DAZ*, *HSFY*, *PRY*, *RBMY*, *TSPY*, *VCY*, *FRG1* and *GLUD1*), as well as four new ones (*FAM47A*, *KRT18*, *TAF13Y* and *TAF11L2*).

**Curation of ampliconic genes.** We first collected annotations from the NCBI annotation pipeline, CAT, and Liftoff. To these annotations, we added mappings from human and species-specific gene sequences onto the latest assemblies and included Iso-seq reads[109] and Iso-seq transcripts[110]. To combine these annotations, we first performed an interval analysis to find all annotated, mapped, or predicted copies, with one or more sources of evidence and then manually curated the final set of protein-coding and pseudogene copies for each of these genes (Supplementary Table 45).

**ddPCR ampliconic gene copy number validations.** Copy numbers were determined with ddPCR using the protocols described[13,39]. The sequences of the primers for bonobo, chimpanzee, gorilla, B. orangutan and S. orangutan were from ref. 39. The primers for siamang were designed using Geneious Prime software[105] and are available in Supplementary Table 34. ddPCR conditions are described in Supplementary Table 35.

**TSPY gene analysis.** The UCSC table browser was used to retrieve and export the *TSPY* sequences. For every genome, the appropriate gene annotation dataset was selected with the specific regions defined using the locations of the curated *TSPY* copies. The sequences of the 5′ UTR, CDS exons, 3′ untranslated regions and introns were retrieved and the generated fasta files were then used for alignment with MAFFT v7.520[111]. Maximum-likelihood phylogenies were inferred using IQTree (v2.0.3)[112] with the best-fit substitution model estimated by ModelFinder[113] (best-fit model according to BIC: TVM + F + G4,

where G4 is G-quadruplexes). Node support values were estimated using 10,000 ultrafast bootstrap replicates[114] with hill-climbing nearest neighbour interchange (−bnni flag) to avoid severe model violations. Nodes with <95% ultrafast bootstrap support were collapsed as polytomies.

**Estimating rDNA copy number and activity by FISH and immuno-FISH.** Chromosome spreads were prepared and labelled as described previously[115]. To estimate rDNA copy number and activity from FISH and Immuno-FISH images, individual rDNA arrays were segmented, the background-subtracted integrated intensity was measured for every array, and the fraction of the total signal of all arrays in a chromosome spread was calculated for each array. Similarly, the fraction of the total UBF fluorescence intensity, indicative of RNA PolI transcription[116], was used to estimate the transcriptional activity of the chrY rDNA arrays. The total rDNA copy number in a genome was estimated from Illumina sequencing data based on *k*-mer counts. Full details are available in Supplementary Methods.

**Gene-level selection using interspecific fixed differences.** To detect selection from interspecific comparison of gene sequences, we started with alignments of ancestral or ampliconic genes, using one consensus sequence per species for ampliconic gene families that were present in at least four species (Supplementary Data 5). For these alignments, we inferred ML phylogeny with raxml-ng (GTR + G + I, default settings otherwise), and looked for evidence of gene-level episodic diversifying selection using BUSTED with site-to-site synonymous rate variation and a flexible random effects branch-site variation for $d_N/d_S$[117,118]. Because all alignments were relatively short, we also fitted the standard MG94 + GTR model where $d_N/d_S$ ratios were constant across sites and were either shared by all branches (global model) or estimated separately for each branch (local model). We tested for $d_N/d_S \neq 1$ using a LRT (global model). To investigate branch-level variability in $d_N/d_S$, we used a version of the local model where all branches except one shared the same $d_N/d_S$ ratio and the focal branch had its own $d_N/d_S$ ratio; *P* values from branch-level $d_N/d_S$ tests were corrected using the Holm–Bonferroni procedure. Finally, to compare mean in global $d_N/d_S$ between ampliconic and ancestral genes, we performed a joint MG94 + GTR model fit to all genes, with the null model that $d_N/d_S$ is the same for all genes, and the alternative model that $d_N/d_S$ are the same within group (ampliconic or ancestral), but different between groups. All analyses were run using[119].

## Methylation analysis

**CpG methylation calling.** To generate CpG methylation calls, Meryl[76] was used to count *k*-mers and compute the 0.02% most frequent 15-mers in each ape draft diploid assembly. ONT and PacBio reads were mapped to the corresponding draft diploid assemblies with Winnowmap2[120] and filtered to remove secondary and unmapped reads. Modbam-2bed (https://github.com/epi2me-labs/modbam2bed) was used to summarize modified base calls and generate a CpG methylation track viewable in IGV[121].

**Methylation analysis.** Using the processed long-read DNA methylation data to analyse large sequence classes (PAR1, Ampliconic regions, ancestral regions), we split these regions into 100-kb bins and calculated mean methylation levels of all CpGs within each bin. For smaller sequence classes, such as specific repetitive elements, we generated mean methylation levels from individual elements themselves. For human data, we added another filtering step to remove regions where two long-read sequencing platforms yielded highly divergent results (mostly Yq12 region); non-human methylation data were concordant between the two sequencing platforms (Supplementary Fig. 18) and thus were used in their entirety. Promoters were defined as regions 1 kb upstream of the transcription start site.

## Diversity analysis

We collected short-read sequencing data from 129 individuals across 11 distinct great ape subspecies (Supplementary Table 42a) and aligned the reads to previous (using the previous reference of S. orangutan reference for B. orangutan data) and T2T sex chromosome assemblies. We next performed variant calling with GATK Haplotype Caller[122], conducted joint genotyping with GenotypeGVCFs[122], and removed low-confident variants. To further enhance the accuracy and completeness of variant detection, we adopted the masking strategy proposed by the T2T-CHM13v2.0 human chrY study[8], in which PARs and/or Y chromosome were masked in a sex-specific manner. After generating karyotype-specific references for XX and XY samples, we realigned the reads of each sample to the updated references and called variants. The new variant set was validated reconstructing the Y chromosome phylogeny and estimating the time-to-most-recent common ancestor on it (Supplementary Note 14). Using the complete variant call sets, we quantified the nucleotide diversity of each subspecies with VCFtools. For chromosome X, we assessed the diversity in PAR and ancestral regions. For chromosome Y, we computed the nucleotide diversity in ancestral regions.

## Reporting summary

Further information on research design is available in the Nature Portfolio Reporting Summary linked to this article.

## Data availability

The raw sequencing data generated in this study have been deposited in the Sequence Read Archive under BioProjects PRJNA602326, PRJNA902025, PRJNA976699, PRJNA976700, PRJNA976701, PRJNA976702, PRJNA986878 and PRJNA986879. The genome assemblies and NCBI annotations are available from GenBank or RefSeq (see Supplementary Table 46 for accession numbers). The CAT/Liftoff annotations are available in a UCSC Genome Browser Hub: https://cgl.gi.ucsc.edu/data/T2T-primates-chrXY/. The reference genomes, alignments and variant calls are also available within the NHGRI AnVIL: https://anvil.terra.bio/#workspaces/anvil-dash-research/AnVIL_Ape_T2T_chrXY. The alignments generated for this project are available at: https://www.bx.psu.edu/makova_lab/data/APE_XY_T2T/ and https://public.gi.ucsc.edu/~hickey/hubs/hub-8-t2t-apes-2023v1/8-t2t-apes-2023v1.hal (with the following additional information: https://public.gi.ucsc.edu/~hickey/hubs/hub-8-t2t-apes-2023v1/8-t2t-apes-2023v1.README.md). Supplementary data include human-specific structural variant coordinates (Supplementary Data 1), sequence class coordinates (Supplementary Data 2), palindrome coordinates (Supplementary Data 3), and RNA-seq and Iso-seq datasets used for gene annotations (Supplementary Data 4), and alignments of ancestral and (consensus) campliconic gene coding sequences (File 5). Primary data related to the cytogenetic evaluation of the rDNA are deposited in the Stowers Institute Original Data Repository under accession LIBPB-2447: https://www.stowers.org/research/publications/libpb-2447 C-values used for genome size estimates (see Supplementary Methods) were taken from the Animal Genome Size Database (https://www.genomesize.com) as found on Genome on a Tree (https://goat.genomehubs.org)[123]. Existing reference assemblies used for comparison can be found under the following accessions on NCBI: GCA_013052645.3 (bonobo, Mhudiblu)[16], GCA_015021855.1 (bonobo; chrY)[15], GCF_002880755.1 (chimpanzee, Clint)[18], GCF_008122165.1 (gorilla, Kamilah)[18], GCA_015021865.1 (gorilla, Jim; chrY)[15], GCA_009914755.4 (human, T2T-CHM13v2.0)[8,20], GCF_002880775.1 (Sumatran orangutan, Suzie)[18] and GCA_015021835.1 (Sumatran orangutan; chrY)[15]. Short-read datasets from other ape individuals used for mapping and diversity analyses were obtained from NCBI under the following accessions: SRP018689[124], ERP001725[56], ERP016782[55] and ERP014340[125] (see Supplementary Table 42).

## Code availability

The source code created to generate the results presented in this paper is publicly available on GitHub (https://github.com/makovalab-psu/T2T_primate_XY) and provided at Zenodo (https://doi.org/10.5281/zenodo.10680008 (ref. 126)). All external scripts and programs are also linked through this GitHub repository.

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

**Acknowledgements** The authors thank R. Campos-Sanchez, S. Canzar, F. Chiaromonte, T. Goldfarb, A. Greshnova, B. de Massy, T. D. Murphy, M. Park, S. Pujar, F. R. Ringeling, C. Steiner, D. J. Taylor, M. Tomaszkiewicz and A. Watwood for their assistance and/or advice; B. Weissensteiner and K. Anthony who assisted with primate cell culture; to PSU Genomics Core Facility, PSU Sartorius Cell Culture Facility and PSU College of Medicine Genome Sciences Core Facility for their technical assistance; and San Diego Zoological Society Frozen Zoo and Tissue and DNA collection, Coriell Institute, Smithsonian Institute, University of Texas MD Anderson Cancer Center and Tulsa Zoo for providing samples and/or cell lines used in this study. This work utilized the computational resources of the NIH HPC Biowulf cluster (https://hpc.nih.gov), and of the Computational Biology Core and sequencing at the Center for Genome Innovation, both in the Institute for Systems Genomics at the University of Connecticut. This work was supported, in part, by the Intramural Research Program of the National Human Genome Research Institute, National Institutes of Health (NIH; to B.D.P, S.N., G.G.B., S.Y.B., A.D., E.P., A.R., S.J.S., A.S., A.C.Y., S.K. and A.M.P.), by the National Center for Biotechnology Information of the National Library of Medicine, NIH (to F.T.-N., D.H., P.M. and K. M. McGarvey), by the NIH awards R01GM130691, R01GM136684, and R35GM151945 (to K.D.M.), HG002385 and HG010169 (to E.E.E.), R01GM146462 (to P. Medvedev), R01CA266339 (to J.L.G.), R01GM123312 (to R.J.O.), R35GM146926 (to Z.A.S.), R35GM146886 (to C.D.H.), R01HG011641 (to S.V.Y.), U01CA253481 and U24HG010263 (to M.C.S.), R35GM124827 (to M.A.W.), R01HG011274 (to K.H.M.), and HG007497 (to C.L. and E.E.E.), by the National Science Foundation awards 2138585 and 1931531 (to P. Medvedev), EF-2204761 (to S.V.Y.), and by the Center for Integration in Science of the Ministry of Aliyah, Israel (I.A.A.). K.H.M. is a Searle Scholar, E.E.E. is an investigator of the Howard Hughes Medical Institute. T.M.L. was supported by the NIH T32 GM102057 Computation, Bioinformatics, and Statistics (CBIOS) Training Program Grant at Penn State University.

**Author contributions** B.D.P. performed computational validations, NCBI submissions, chimpanzee subspecies identification, biosample registration, figure generation and overall project and consortium coordination. R.S.H. generated alignments, identified pseudoautosomal boundaries and palindromes, including their sharing, and performed substitution analysis. M.C. classified assemblies into sequence classes, identified ampliconic regions, and performed palindrome analysis. G.A.H., P.G.S.G., J.M.S., R.J.O. and S.J.H. performed repetitive element annotation, manual curation, analyses and dfam submissions. J.M.S. performed lineage-specific repeat analyses. G.A.H. generated tracks for figures. K.P. performed gene density analyses, visualized palindrome sharing, and identified multi-copy and ampliconic gene families. S.N. and S.K. performed sequence assemblies. G.H. and B.P. generated multi-species alignments. A.S. generated dot plots. S.J.S. performed rDNA array copy number estimation, base calling and alignment, and generated methylation tracks. D.Y., W.T.H. and H.J. performed segmental duplication and structural variation analyses. D.Y. also identified percentages of chromosomes aligned to each other. Q.L., A.B., M.C.S., R.C.M., M.G.T., C.D.H., T.M.L., S., Z.A.S., P. Hallast, C.L. and S.L.K.P. performed diversity and selection analyses. K.P., P. Hebbar, F.T.-N., D.H., P. Masterson, M.A.W., B.J.P., M.G.T. and M.D. performed gene annotations and analyses. K.K. performed non-B-DNA analysis. X.Z. performed substitution spectrum analysis, collected species divergence times from the literature and assisted in figure preparation. D.E.C., K.S., P.-C.C. and A.C. performed DeepConsensus calling. M.A. and E.B.-B. performed de novo gene analysis. C.S.-C. analysed palindrome structure in orangutans. P.H.S. and J.L.R. provided HiFi data for bonobo. I.A.A., F.R., V.A.S., V.S. and K.H.M. performed centromere analysis. S.V.Y., D.A.H. and Y.-H.E.L. performed methylation analysis. T.P., M.B. and J.L.G. performed rDNA analysis. A.D. and E.P. generated karyotypes. G.A.H., L. Carbone and R.J.O. confirmed the siamang karyotype. L.d.G. and M.V. performed karyotype confirmation and FISH analysis on rDNA. H.Z. performed ddPCR and maintained cell culture. A.C.Y., S.Y.B. and G.G.B. generated UL-ONT and Illumina sequences. S.S. and R.E.G. generated Hi-C libraries. K. M. Munson, A.P.L. and G.H.G. generated HiFi and Iso-seq PacBio sequences. A.R., P.M. and S.J.C.C. participated in project discussions, S.J.C.C. also collected gene ontology and mating system information, and A.R. performed methylation comparison between two sequencing platforms. L. Carrel, L. Carbone and O.A.R. provided samples. L. Carbone also provided karyotype confirmation. B.C.M. coordinated project resources, maintained cell culture, and performed ddPCR and RNA extractions. K.D.M., E.E.E. and A.M.P. provided project leadership and coordination, and are co-leading the Primate T2T Consortium. K.D.M. wrote the manuscript with contributions from the other authors.

**Competing interests** E.E.E. is a member of the scientific advisory board of Variant Bio. R.J.O. is a member of the scientific advisory board of Colossal Biosciences. C.L. is a member of the scientific advisory boards of Nabsys and Genome Insight. The other authors declare no competing interests.

**Additional information**
**Correspondence and requests for materials** should be addressed to Kateryna D. Makova, Evan E. Eichler or Adam M. Phillippy.

**a** Chromosome X

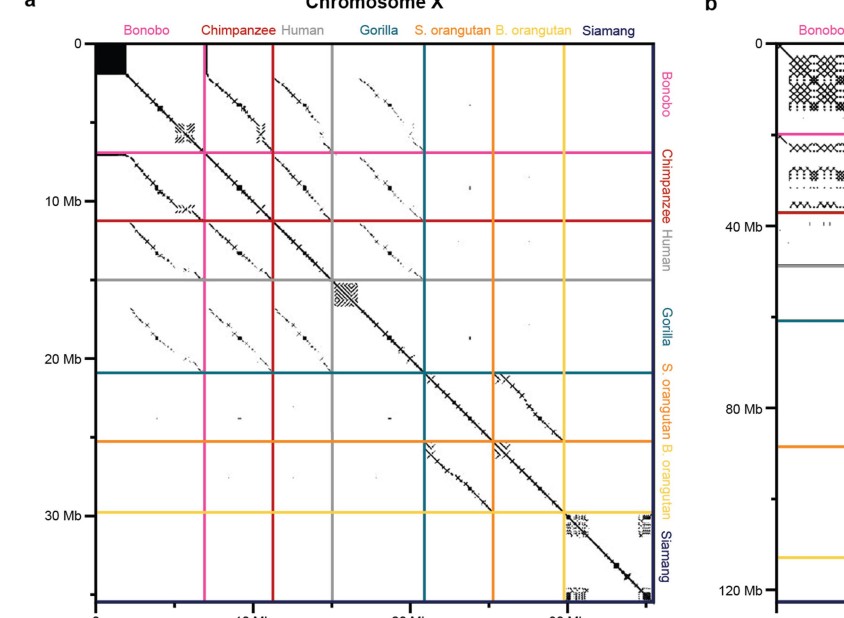

**b** Chromosome Y

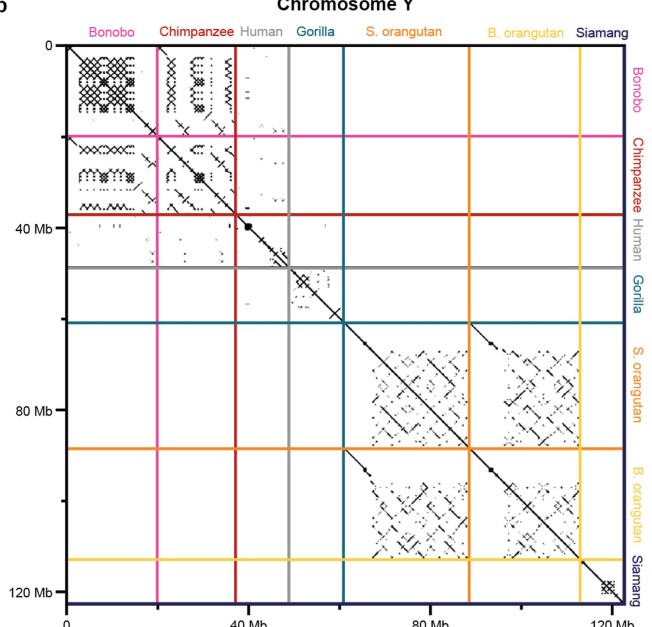

**Extended Data Fig. 1 | Conservation of ampliconic regions across species.**
A between-species comparison of ampliconic regions on the (**a**) X chromosomes and (**b**) Y chromosomes between species with similarities highlighted using a dot plot analysis. Ampliconic regions were extracted and concatenated independently for each species and visualized with gepard[127] using a window size of 100.

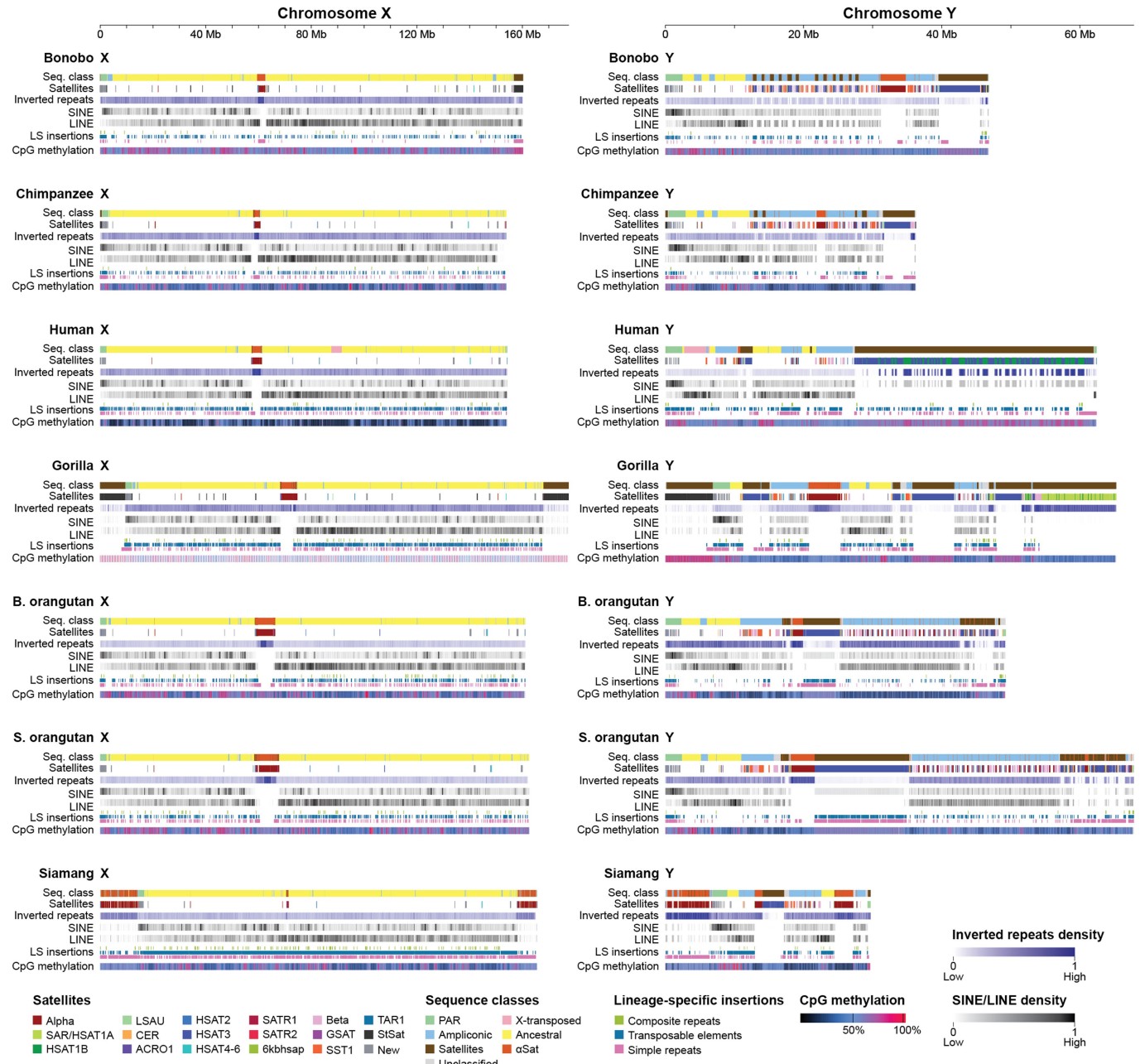

**Extended Data Fig. 2 | Repeats and satellites on the X and Y chromosomes.**
Repeats and satellites shown with sequence class annotations and CpG methylation for chromosomes X and Y. The scales are different between chromosomes X and Y. The tracks for each species are: (1) sequence class annotation, (2) satellites, (3) inverted repeats, (4) SINEs, (5) LINEs, (6) lineage-specific (LS) insertions of composite repeats (green), transposable elements (blue), and satellites, simple repeats, and low-complexity repeats (pink), and (7) CpG methylation. The inverted repeat, SINE, and LINE tracks are plotted in blocks with darker colors representing a higher density (density values are calibrated independently for each chromosome/species). CpG methylation is also displayed on a gradient between dark blue (low methylation) and magenta (high methylation) based on the percentage of supporting aligned ONT reads. The remaining tracks (sequence class, satellites, and LS insertions) are displayed as presence/absence (color/no color). The class and satellite tracks are discrete, whereas the LS insertions are plotted as mini tracks to avoid overplotting where >1 label applies.

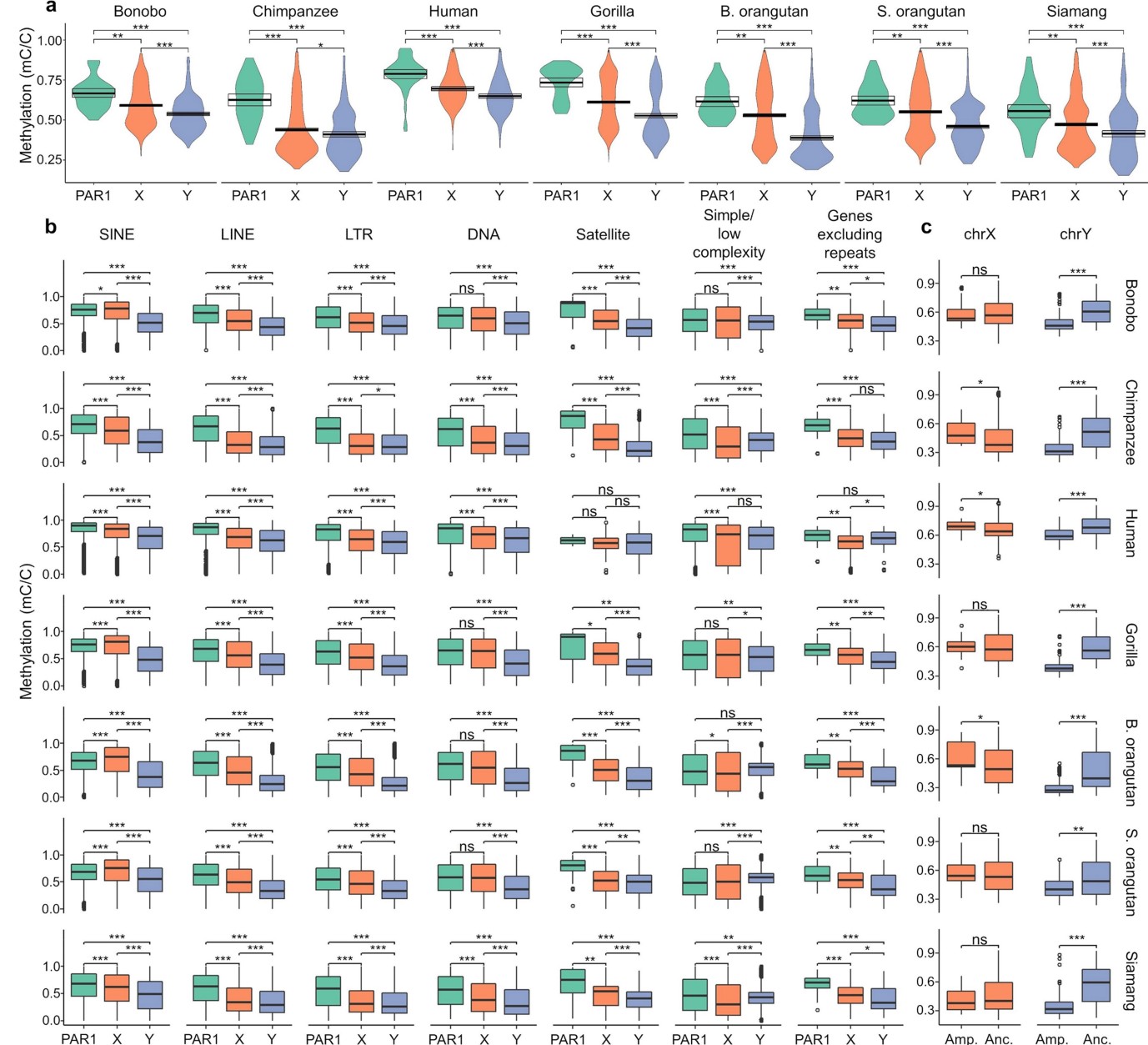

**Extended Data Fig. 3 | Methylation patterns.** (**a**) DNA methylation levels in 100-kb bins in Pseudoautosomal region 1 (PAR1; teal), non-PAR chromosome X (orange), and non-PAR chromosome Y (periwinkle). (**b**) Differences in DNA methylation levels between different repeat categories as well as protein-coding genes (after excluding repetitive sequences). (**c**) Differences in methylation levels between ampliconic and ancestral regions in the X and the Y chromosomes (in 100 kb bins). All box plots (**a-c**) show the median and first and third quartiles.

Those in **b-c** also have whiskers extending to the closer of the minimum/maximum value or 1.5 times the interquartile range, and outliers (beyond the whiskers) are plotted as individual points. $p$-values were determined using two-sided Wilcoxon rank-sum tests (* $p < 0.05$; ** $p < 10^{-3}$; *** $p < 10^{-6}$) and are shown in Table S28. No correction for multiple testing was applied. Sample sizes (i.e., number of 100 kb bins (a,c) or number of repeats, genes, etc. (b)) are shown in Table S28.

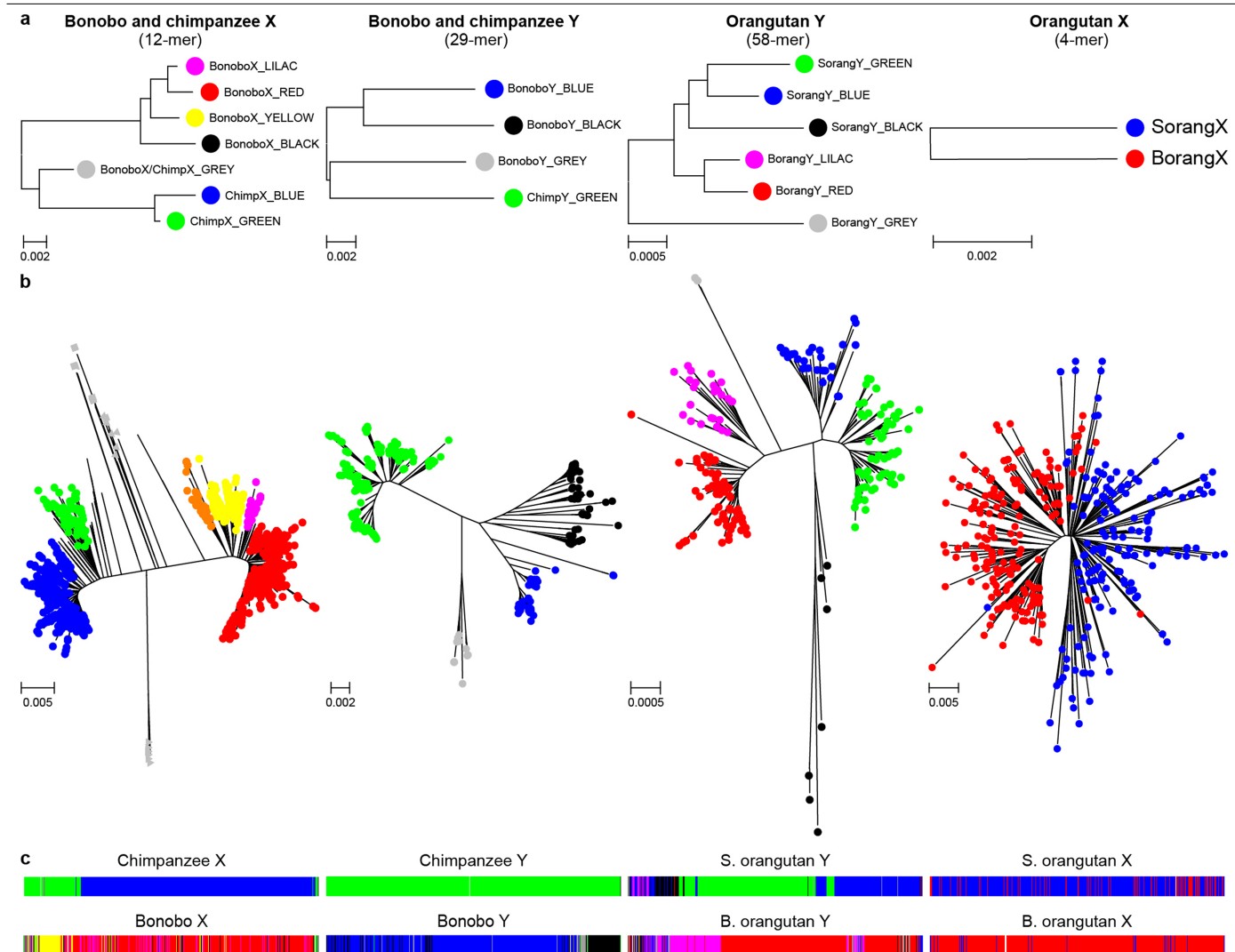

**Extended Data Fig. 4 | Alpha satellite higher order repeat (HOR) haplotypes are species-specific in *Pongo* and *Pan* (except for the few distal HOR copies).**
**(a)** Consensus HOR haplotype (HORhap) phylogenetic trees, **(b)** HOR trees, and **(c)** HORhap UCSC Genome Browser annotation tracks for active alpha satellite arrays of chromosomes X and Y in two *Pan* and two *Pongo* species (see Methods in Note S7) are shown. Each colored branch in a HOR tree represents a HORhap. All branches in HOR trees are species-specific, except for the GREY cluster in *Pan* cenX tree, where mixing of chimpanzee (square markers) and bonobo (triangle markers) HORs were observed (Note S7). Each branch was extracted to obtain HORhap consensus sequence and HMM further used in HMMER-based HORhap classification tool[44] to produce HORhap annotations. The larger branches with shorter twigs correspond to the younger large active HORhap arrays; the smaller branches with longer twigs correspond to the older and smaller side arrays. The thinnest and longest branches make up the oldest and

smallest peripheral arrays which often cannot be seen in the track panels. The *Pongo* X tree has a 'star-like' shape and does not have obvious HORhaps; HORs colored by species indicate almost no mixing between species and species-specific consensus sequences show three consistent differences (Fig. S15D). Thus, we concluded that the species did not share the same HORhaps, but no significant divides could be seen in the tree due to the short HOR length (a 4-mer), as detailed in Note S7. The age of the HORhaps is also confirmed by consensus trees where the oldest GREY twigs branch out closer to the root and are nearly equidistant to the active HORhap branches of respective species. Hence they likely resemble the HORs that existed in the common ancestor of both species. Thus, all but the oldest HORhaps are species-specific and indicate considerable evolution that occurred after the species diverged.

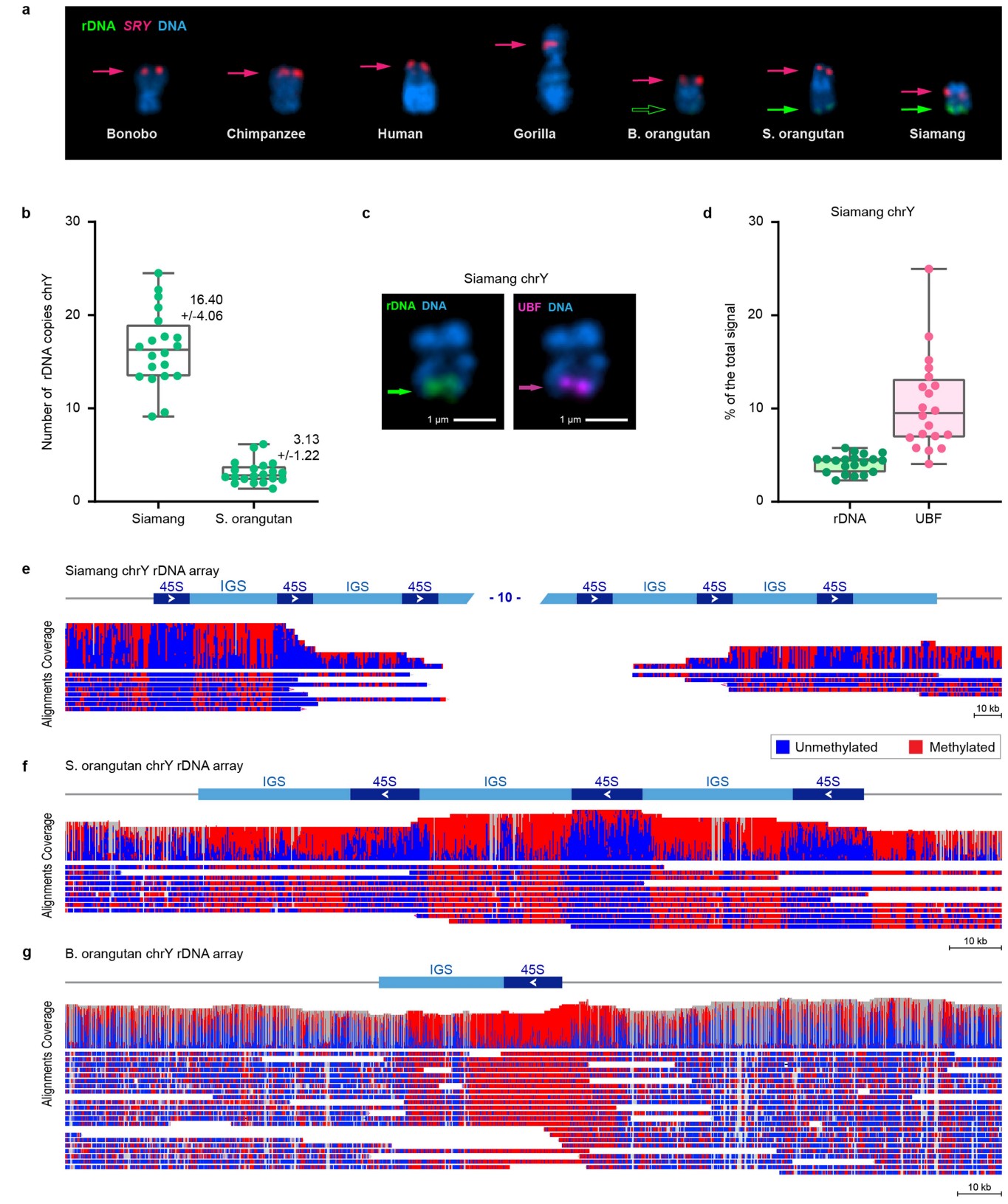

**Extended Data Fig. 5** | See next page for caption.

**Extended Data Fig. 5 | Estimation of rDNA copy number and activity on chromosome Y arrays.** (**a**) Gallery view of Y chromosomes from species in this study. Chromosomes were FISH-labeled with rDNA- (BAC RP11-450E20, green) and *SRY*-containing (BAC RP11-400O10, red) BAC probes and counter-stained with DAPI. Siamang and both orangutans' Y chromosomes have rDNA signal on the distal ends of the q-arms. (**b**) Siamang and Sumatran orangutan chrY rDNA copy number was quantified from the fraction of the total fluorescent intensity of rDNA signals on all chromosomes (from chromosome spreads as in panel **a**) and the Illumina sequencing estimate of the total copy number of rDNA repeats in the genome (339 copies in siamang, 814 in Sumatran orangutan). The mean and standard deviations from 20 chromosome spreads are shown near each box plot. The box plots show the median as the center line and the first and third quartiles as the bounds of the box; the whiskers extend to the minimum/maximum value, and all values are plotted as dots in front of the box plot. The rounded average of rDNA arrays on chrY were 16 copies for siamang and 3 copies for Sumatran orangutan. (**c**) A representative image of siamang chrY labeled by immuno-FISH with rDNA probe (green) and the antibody against rDNA transcription factor UBF (magenta). The chrY rDNA array is positive for the UBF signal. (**d**) Quantification of siamang chrY rDNA and UBF expressed as the fraction of the total fluorescent intensity of all rDNA-containing chromosomes in a chromosome spread. The box plots are plotted as in **b** from 20 chromosome spreads. ChrY rDNA arrays contain on average ~10% of the total chromosomal UBF signal. Siamang (**e**) and Sumatran (**f**) and Bornean (**g**) orangutan read-level plots showing ONT methylation patterns at the chrY rDNA locus and surrounding regions. The coverage track shows the depth of sequencing coverage across the rDNA array, and the methylation track displays the methylation status of individual cytosines. Hypomethylation of the 45 S units is evidence of active transcription in siamang and S. orangutan, but not B. orangutan. Only reads >100 kb that are anchored in unique sequence outside the rDNA array and (except for Bornean orangutan) span at least two 45 S units are shown.

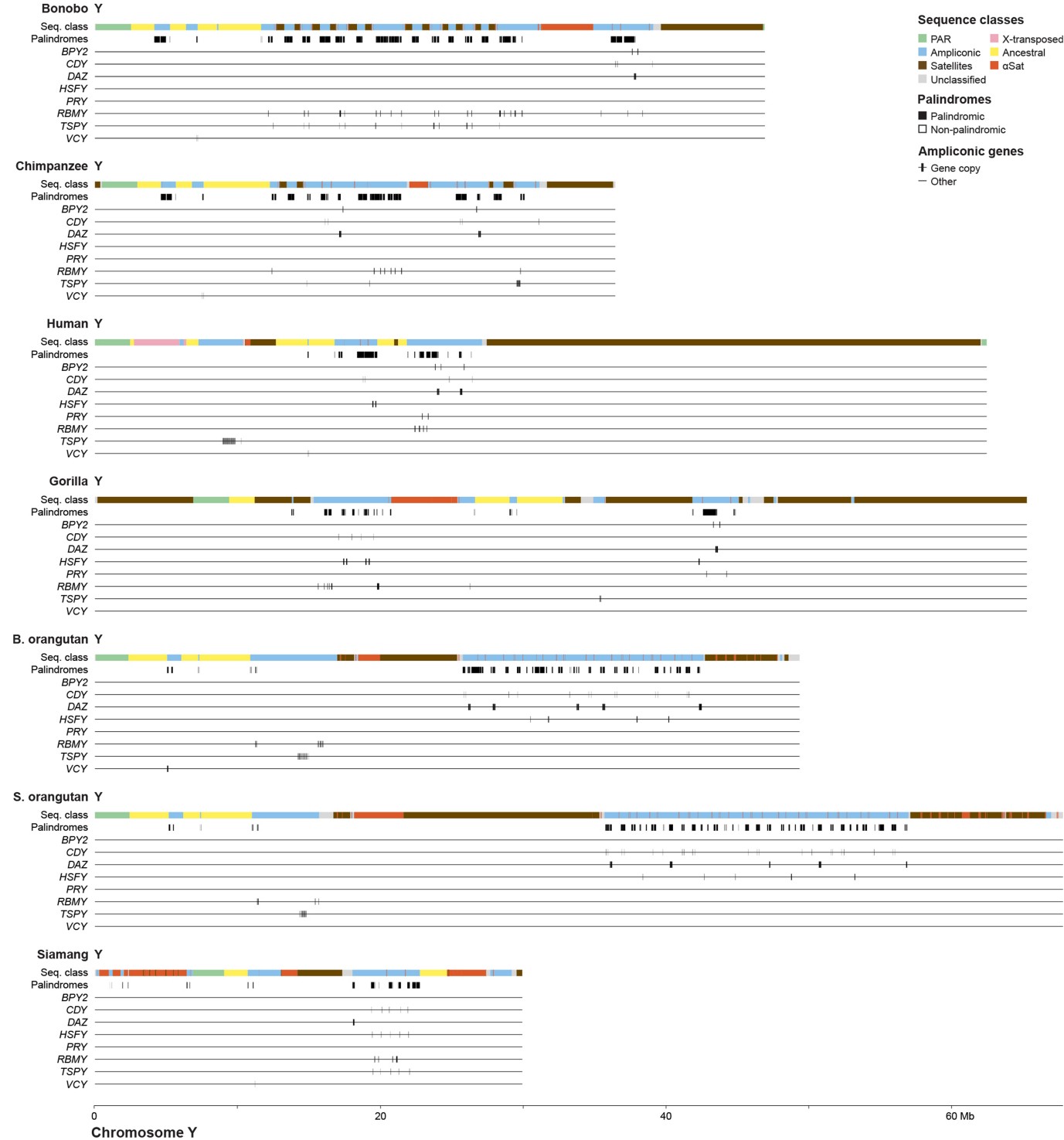

**Extended Data Fig. 6 | Positions of ampliconic gene families on the Y chromosome.** Locations of protein-coding ampliconic genes, grouped by family, are shown with sequence class annotations and palindrome locations on each Y chromosome. The tracks for each species are: (1) sequence class annotation, (2) palindromes, and (3-9) ampliconic gene families: *BPY2, CDY,* *DAZ, HSFY, PRY, RBMY, TSPY,* and *VCY.* The sequence class track has a discrete class annotation for every base. All other tracks are displayed as presence/ absence (color/no color) with the ampliconic gene family tracks containing a horizontal midline to help the eye with the sparse display. All Y chromosomes are plotted on the same scale.

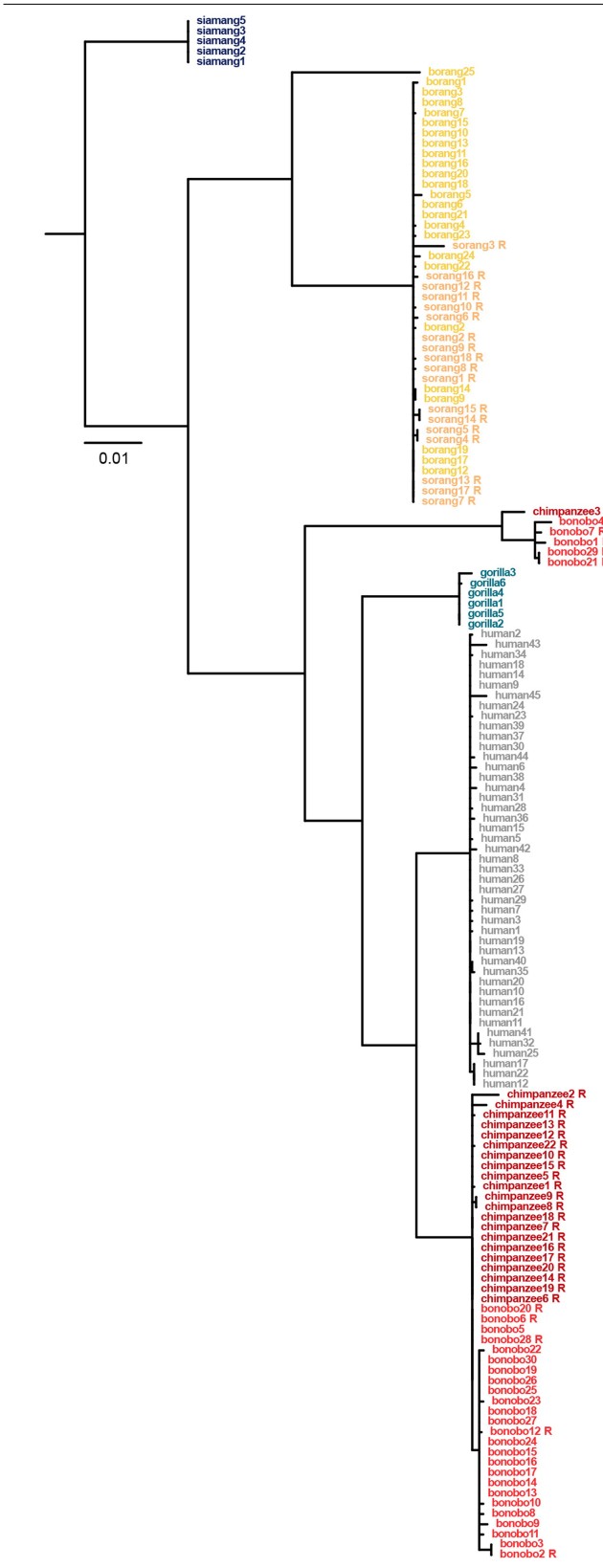

**Extended Data Fig. 7 | Phylogenetic analysis of the *TSPY* gene family.**
Phylogenetic analysis (see Methods) of the protein-coding copies of the *TSPY* gene family in great apes, using siamang as an outgroup, uncovered mostly lineage-specific clustering suggesting homogenization among copies. Gene copies (numbered for each species) were extracted from the manually curated set (Table S45) and included 5′ and 3′ UTRs, CDS exons, and introns. These sequences were aligned and used to infer a maximum likelihood phylogeny (see Methods for details) with 10,000 ultrafast bootstrap replicates. Nodes with <95% bootstrap support were collapsed. 'R' indicated a reverse orientation as compared with the assembly sequences.

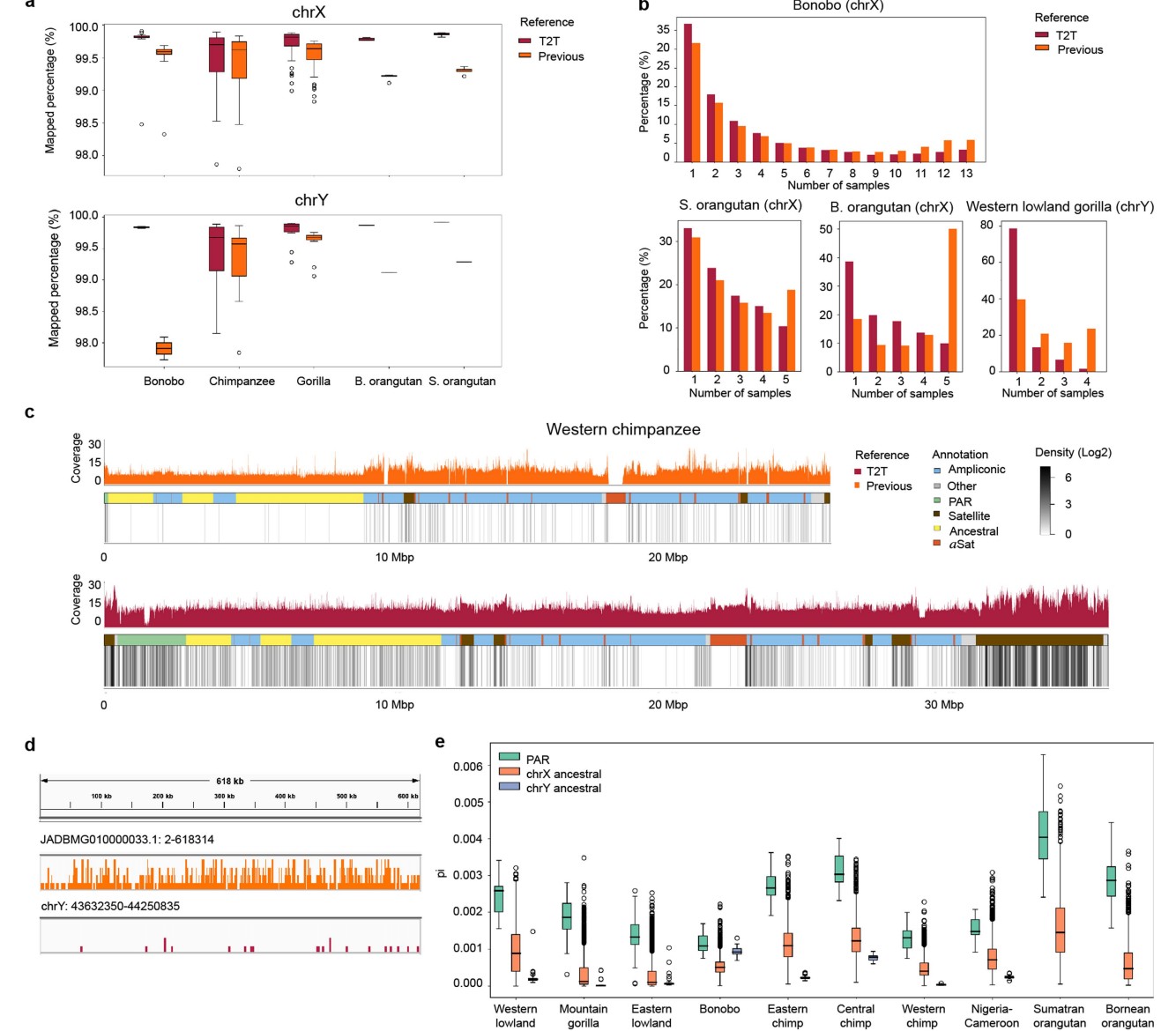

**Extended Data Fig. 8 | T2T assemblies facilitate short-read mapping and enable the analysis of genetic diversity in great apes. (a)** The percentage of short reads mapped to T2T vs. previous sex chromosome assemblies (using the previous reference assembly of Sumatran orangutan for Bornean orangutan data). Reads were sourced from multiple individuals per species, and the number of individuals per species and the total number of reads per species (sum of reads per individual) are listed in Table S42. The box plots show the median as the center line and the first and third quartiles as the bounds of the box; the whiskers extend to the closer of the minimum/maximum value or 1.5 times the interquartile range. Outliers (beyond the whiskers) are plotted as individual points. **(b)** Allele frequencies (y-axis) of variants called from reads mapped to T2T vs. previous assemblies. **(c)** Coverage and variant density (in log2 values of densities per 10 kb) distribution across previous (shown in the reverse orientation) and T2T assemblies for western chimpanzee. Peak variant densities were observed at 5.9 for previous chrY and at 7.6 for T2T chrY. **(d)** Distributions of variant allele frequencies on JADBMG010000033.1 (positions 2 to 618,314, upper), a contig from a previous chrY assembly, and T2T chrY (positions 43,632,350 to 44,250,835, bottom), for western lowland gorilla, visualized using IGV. **(e)** Nucleotide diversity (pi)[128] in pseudoautosomal regions (PARs), ancestral regions of chromosome X, and ancestral regions of chromosome Y. 'Chimp' stands for chimpanzee. Variants for the calculation of pi were called for multiple individuals per subspecies, and the number of individuals per subspecies and the total number of variants per region (sum of variants per individual) are listed in Table S42. The box plots were plotted as in **a**.

# Reporting Summary

## Statistics

For all statistical analyses, confirm that the following items are present in the figure legend, table legend, main text, or Methods section.

| n/a | Confirmed | |
|---|---|---|
| ☐ | ☒ | The exact sample size (*n*) for each experimental group/condition, given as a discrete number and unit of measurement |
| ☐ | ☒ | A statement on whether measurements were taken from distinct samples or whether the same sample was measured repeatedly |
| ☐ | ☒ | The statistical test(s) used AND whether they are one- or two-sided *Only common tests should be described solely by name; describe more complex techniques in the Methods section.* |
| ☒ | ☐ | A description of all covariates tested |
| ☐ | ☒ | A description of any assumptions or corrections, such as tests of normality and adjustment for multiple comparisons |
| ☐ | ☒ | A full description of the statistical parameters including central tendency (e.g. means) or other basic estimates (e.g. regression coefficient) AND variation (e.g. standard deviation) or associated estimates of uncertainty (e.g. confidence intervals) |
| ☐ | ☒ | For null hypothesis testing, the test statistic (e.g. *F*, *t*, *r*) with confidence intervals, effect sizes, degrees of freedom and *P* value noted *Give P values as exact values whenever suitable.* |
| ☒ | ☐ | For Bayesian analysis, information on the choice of priors and Markov chain Monte Carlo settings |
| ☒ | ☐ | For hierarchical and complex designs, identification of the appropriate level for tests and full reporting of outcomes |
| ☐ | ☒ | Estimates of effect sizes (e.g. Cohen's *d*, Pearson's *r*), indicating how they were calculated |

*Our web collection on statistics for biologists contains articles on many of the points above.*

## Software and code

Policy information about availability of computer code

| | |
|---|---|
| Data collection | Raw data collection relied on software installed on various instruments (e.g., a sequencing machine), which would be available to any user of such an instrument, and, when relevant, we have noted version numbers of such software packages in the text alongside the information on instrumentation and other methods. No new data collection software was created for this work, though many mundane data organization and sharing tasks (e.g., renaming and group files or sharing data with collaborators over the internet) were performed that were not documented because they are relatively commonplace and uninteresting. All subsequent processing of raw data is considered part of the "analysis" of the data, and all analyses are extensively documented in the text (especially in the methods and supplementary information). Please see our response in the "Data analysis" portion of the "Software and code" availability of this form for more detail. |
| Data analysis | Analyses primarily relied on previously-published, open-source software, and such software packages are mentioned in the text (especially the methods and supplementary information), with a version number and citation. No new algorithms or software packages were created for this study, though some custom scripts were created and run for certain tasks. When the methods for a particular analysis would be difficult to replicate from the text with listed options and/or commands alone, more detailed code is provided in a publicly-available repository on GitHub (https://github.com/makovalab-psu/T2T_primate_XY) and at Zenodo (https://doi.org/10.5281/zenodo.10680008). As necessary, external scripts and programs are also linked through this GitHub repository. All tools and versions from paper listed below: Counting rDNA Copy Number analysis on GitHub: jellyfish 2.2.10 samtools 1.3.1 |

htslib 1.3.1
bedtools v2.29.0
Denovo genes
Github
Python=3.8.10

Python packages:
Bio=1.81,
pandas=1.5.3

methylation analysis on GitHub:
R version 4.3.2
R packages:
cowplot=1.1.3
GenomicRanges=1.54.1
ggplot2=3.5.0,
ggsignif=0.6.4,
scales=1.3.0

Multicopy genes analysis on GitHub:
Python 3.12.2

Python packages:
bcbio-gff==0.7.0
biopython==1.81
gspread==5.9.0
pandas==2.0.1
seaborn==0.12.2
matplotlib==3.7.1

palindromes analysis on GitHub:
Conda environment:
name: alignment
channels:
  - anaconda
  - bioconda
  - defaults
dependencies:
  - _libgcc_mutex=0.1
  - _openmp_mutex=5.1
  - bamtools=2.5.1
  - bedtools=2.30.0
  - bioawk=1.0
  - blas=1.0
  - blast=2.5.0
  - blat=35
  - boost=1.73.0
  - bzip2=1.0.8
  - c-ares=1.19.0
  - ca-certificates=2023.01.10
  - curl=7.88.1
  - emboss=6.6.0
  - expat=2.4.9
  - fasta-splitter=0.2.6
  - fontconfig=2.14.1
  - freetype=2.12.1
  - giflib=5.2.1
  - gsl=2.6
  - icu=58.2
  - jemalloc=5.2.1
  - jpeg=9e
  - k8=0.2.5
  - krb5=1.19.4
  - ld_impl_linux-64=2.38
  - lerc=3.0
  - libboost=1.73.0
  - libcurl=7.88.1
  - libdeflate=1.17
  - libedit=3.1.20221030
  - libev=4.33
  - libffi=3.4.4
  - libgcc-ng=11.2.0
  - libgd=2.3.3
  - libgfortran-ng=11.2.0
  - libgfortran5=11.2.0
  - libgomp=11.2.0

- libnghttp2=1.46.0
- libnsl=2.0.0
- libopenblas=0.3.21
- libpng=1.6.39
- libssh2=1.10.0
- libstdcxx-ng=11.2.0
- libtiff=4.5.0
- libuuid=1.41.5
- libwebp=1.2.4
- libwebp-base=1.2.4
- libxml2=2.10.3
- lz4-c=1.9.4
- mashmap=2.0
- meryl=1.3
- minimap2=2.17
- ncurses=6.4
- nomkl=3.0
- numpy=1.23.5
- numpy-base=1.23.5
- openblas=0.3.21
- openblas-devel=0.3.21
- openssl=1.1.1u
- perl=5.32.1
- perl-constant=1.33
- perl-exporter=5.72
- perl-file-util=4.201720
- perl-lib=0.63
- pip=23.0.1
- py-boost=1.73.0
- python=3.10.11
- readline=8.2
- samtools=1.6
- seqtk=1.3
- setuptools=67.8.0
- sqlite=3.41.2
- tk=8.6.12
- tzdata=2023c
- wfmash=0.7.0
- wheel=0.38.4
- winnowmap=2.03
- xz=5.2.10
- zlib=1.2.13
- zstd=1.5.5

palindrover_maf_align analysis on GitHub:
Python 3.11.0+
Lastz 1.04.22

Python packages
palindrover 0.1.5 (included)
maf_alignments 0.1.0

Phylogenetic inference analysis on GitHub:
Python=3.9.11

Python packages:
Bio=1.79
pandas=1.4.1
numpy=1.21.2

Sequence classes analysis on Github:
Bedtools=v2.30.0
Samtools=1.6
Python=3.10.11
Blast=2.5.0+
Circos=v 0.69-8 | 15 Jun 2019 | Perl 5.032001
xtr_search
Github
R=4.3.0
R packages
ggplot=3.4.3
cowplot=1.1.1

Sequencing analysis in Supplementary methods:
Guppy=6.0.0+

Assembly
Suppl.
methods
Verrko v1.1
Bandage v0.8.1
BangageNG v2022.09
IGV v2.25.4
Merqury v1.3
Hifiasm v0.16.1-r375
MashMap v2.0
Flye=v2.9-b1768
BWA-MEM2 v2.2.1
Winnowmap2 v2.03
Meryl v1.03
Non-B annot
Suppl.
methods
non-B_gfa (commit 2f5a24f)
Alignments
Suppl.
methods
Minimap2 2.24
Lastz 1.04.22

Phylogenetic analysis in Supplementary methods:
Cactus 2.6.0
Hal2maf 2.2
IQTree 2.0.3

Substitution Frequency analysis in Supplementary methods:
Cactus 2.2.0
Maftools Oct 20 2022
phast 1.5

Gene annotations (NCBI) analysis in Supplementary methods:
BUSCO 4.0.2

Gene annotation (UCSC) analysis in Supplementary methods:
Cactus 2.6.0
Minimap2
CAT 2.2.1
Liftoff 1.6.3

Repeat and satellite annotation analysis in Supplementary methods:
RepeatMaske v4.1.2-p1
Dfam 3.6 [data]
Repbase v20181026
Bedtools 2.29.0
TRF v4.09
ULTRA v1.0
Minidot v2016
Rideogram v0.2.2

Sequence Classes analysis in Supplementary methods:
SEDEF v1.1
TRF v4.0.9
RepeateMasker v4.1.2-p1
Windowmaser v2.2.22
Blast 2.5.0+
palindrover  20230615

Segmental duplications analysis in Supplementary methods:
SEDEF v1.1
TRF v4.0.9
Windowmasker v2.2.22
Minimap2 2.24
Blast 2.5.0+
SV's
Suppl.
methods
Minimap2 2.24
Bedtools 2.29.2
Rustybam v0.1.29

rDNA arrays analysis in Supplementary methods:
Ribotin=v1.1.0

T2T-Polish=v1.0

Methylation analysis in Supplementary methods:
Meryl v1.3
Guppy v6.3.8
Pbccs v6.4.0
Primrose v1.3.0
Winnowmapt v2.03
Samtools 1.17
ModBam2bed=v0.6.2
featureCounts v2.0.6

Diversity analysis in Supplementary methods:
BWA-MEM v0.7.17
GATK v4.4.0.0
VCFtools v0.1.16

Y phylogeny TMRCA analysis in Supplementary methods:
VCFtoold v0.1.16
BEAST v1.10.4
IQ-TREE 1.6.12
Tree-Annotator 1.10.4
FigTree v1.4.4

Chimp subspecies identification analysis in Supplementary notes S2:
Seqtk v1.4
Mash (commit 41ddc61)
Clustal Omega v1.2.4
Bowtie2 v2.5.1
Samtools v1.17
Mashmap v2.0
tRNAscan-SE v2.0.11
nucmer v4.0.0rc1

Bonobo PAR2 and Ariel satellites analysis in Supplementary notes S3:
BLAST+ v2.14.0
Winnowmap v2.03
SAMtools v1.18
Mashmap v2.0

XTR analysis in Supplementary notes S4:
R v4.3.0
ggplot2 v3.4.3
cowplot v1.1.1

De novo genes analysis in Supplementary notes S11:
BLAST 2.12.0 (accessed online Oct 2023)
NetSurfP-3.0
fIDPnn (has no version number)
AGGRESCAN (has no version number)
Protein-Sol (has no version number)
ESMFold (has no version number)
All tools used/accessed in October 2023)

Selection analysis of ancestral (X-degenerate) Y genes using diversity data in Supplementary notes S13:
Bcftools 1.9
CACTUS 3.1.20211107152837
vcfR 1.14.0
Ape 5.7
R 4.3.0
Hal2maf 2.2
IQTree 2.0.3
AGAT 1.2.0
MACSE 2.07
HyPhy 2.5.50
Gblocks 0.91b
aBSREL 2.23.0

For manuscripts utilizing custom algorithms or software that are central to the research but not yet described in published literature, software must be made available to editors and reviewers. We strongly encourage code deposition in a community repository (e.g. GitHub). See the Nature Portfolio guidelines for submitting code & software for further information.

## Data

The raw sequencing data generated in this study have been deposited in the Sequence Read Archive under BioProjects PRJNA602326, PRJNA902025, PRJNA976699-PRJNA976702, and PRJNA986878-PRJNA986879. The genome assemblies and NCBI annotations are available from GenBank/RefSeq (see Table S46 for accession numbers). The CAT/Liftoff annotations are available in a UCSC Genome Browser Hub: https://cgl.gi.ucsc.edu/data/T2T-primates-chrXY/. The reference genomes, alignments and variant calls are also available within the NHGRI AnVIL: https://anvil.terra.bio/#workspaces/anvil-dash-research/AnVIL_Ape_T2T_chrXY. The alignments generated for this project are available at: https://www.bx.psu.edu/makova_lab/data/APE_XY_T2T/ and https://public.gi.ucsc.edu/~hickey/hubs/hub-8-t2t-apes-2023v1/8-t2t-apes-2023v1.hal (with the following additional information: https://public.gi.ucsc.edu/~hickey/hubs/hub-8-t2t-apes-2023v1/8-t2t-apes-2023v1.README.md)  Additional Data Files include human-specific structural variant coordinates (File 1), sequence class coordinates (File 2), palindrome coordinates (File 3), RNA-Seq and IsoSeq datasets used for gene annotations (File 4), and manual annotations of Y ampliconic genes (File 5).  Primary data related to the cytogenetic evaluation of the rDNA is deposited in the Stowers Institute Original Data Repository under accession LIBPB-2447: https://www.stowers.org/research/publications/libpb-2447. C-values used for genome size estimates (see Supplemental Methods) were taken from the Animal Genome Size Database (https://www.genomesize.com) as found on Genome on a Tree (https://goat.genomehubs.org). Existing reference assemblies used for comparison can be found under the following accessions on NCBI: GCA_013052645.3 (bonobo, Mhudiblu), GCA_015021855.1 (bonobo; chrY), GCF_002880755.1 (chimpanzee, Clint), GCF_008122165.1 (gorilla, Kamilah), GCA_015021865.1 (gorilla, Jim; chrY), GCA_009914755.4 (human, T2T-CHM13v2.0), GCF_002880775.1 (Sumatran orangutan, Suzie), and GCA_015021835.1 (Sumatran orangutan; chrY). Short read datasets from other ape individuals used for mapping and diversity analyses were obtained from NCBI under the following accessions: SRP018689, ERP001725, ERP016782, and ERP014340 (see Table S42).

## Research involving human participants, their data, or biological material

| Reporting on sex and gender | N/A: No human participants were part of this study |
|---|---|
| Reporting on race, ethnicity, or other socially relevant groupings | *Please specify the socially constructed or socially relevant categorization variable(s) used in your manuscript and explain why they were used. Please note that such variables should not be used as proxies for other socially constructed/relevant variables (for example, race or ethnicity should not be used as a proxy for socioeconomic status). Provide clear definitions of the relevant terms used, how they were provided (by the participants/respondents, the researchers, or third parties), and the method(s) used to classify people into the different categories (e.g. self-report, census or administrative data, social media data, etc.) Please provide details about how you controlled for confounding variables in your analyses.* |
| Population characteristics | *Describe the covariate-relevant population characteristics of the human research participants (e.g. age, genotypic information, past and current diagnosis and treatment categories). If you filled out the behavioural & social sciences study design questions and have nothing to add here, write "See above."* |
| Recruitment | *Describe how participants were recruited. Outline any potential self-selection bias or other biases that may be present and how these are likely to impact results.* |
| Ethics oversight | *Identify the organization(s) that approved the study protocol.* |

Note that full information on the approval of the study protocol must also be provided in the manuscript.

# Field-specific reporting

Please select the one below that is the best fit for your research. If you are not sure, read the appropriate sections before making your selection.

☒ Life sciences ☐ Behavioural & social sciences ☐ Ecological, evolutionary & environmental sciences

For a reference copy of the document with all sections, see nature.com/documents/nr-reporting-summary-flat.pdf

# Life sciences study design

All studies must disclose on these points even when the disclosure is negative.

| Sample size | We analyzed one cell line per species as an initial investigation of sex chromosomes in great apes. No sample size calculation was done. |
|---|---|
| Data exclusions | No data were excluded from the analyses |
| Replication | We did not replicate our experiments except for (1) droplet digital PCR that was conducted in triplicates, and (2) FISH experiments for rDNA probing (conducted in 20 replicates). All attempts at replication were successful. Most additional analyses were computational and thus were |

| | not conducted in replicates. |
|---|---|
| Randomization | N/A because we only used one sample per species |
| Blinding | N/A because we only used one sample per species |

# Reporting for specific materials, systems and methods

We require information from authors about some types of materials, experimental systems and methods used in many studies. Here, indicate whether each material, system or method listed is relevant to your study. If you are not sure if a list item applies to your research, read the appropriate section before selecting a response.

## Materials & experimental systems

| n/a | Involved in the study |
|---|---|
| ☐ | ☒ Antibodies |
| ☐ | ☒ Eukaryotic cell lines |
| ☒ | ☐ Palaeontology and archaeology |
| ☒ | ☐ Animals and other organisms |
| ☒ | ☐ Clinical data |
| ☒ | ☐ Dual use research of concern |
| ☒ | ☐ Plants |

## Methods

| n/a | Involved in the study |
|---|---|
| ☒ | ☐ ChIP-seq |
| ☒ | ☐ Flow cytometry |
| ☒ | ☐ MRI-based neuroimaging |

## Antibodies

| Antibodies used | 1. rabbit polyclonal anti-UBF, Novus Biologicals, cat.# NBP1-82545<br>2. goat anti-rabbit Alexa Fluor 647, Thermo Fisher Scientific |
|---|---|
| Validation | 1. Novus Biologicals website lists seven publications using this antibody, including one publication that uses it for immunofluorescence.<br>2. Thermo Fisher's website lists many publications using this antibody. The website also specifies that "the sensitivity and specificity of each lot is confirmed using ELISA" and there is "minimal cross-reactivity with mouse, rat, human, bovine, guinea pig, and donkey IgG". |

## Eukaryotic cell lines

Policy information about cell lines and Sex and Gender in Research

| Cell line source(s) | KB8711 (San Diego Zoological Society), AG18354 (Coriell), AG06213 (Coriell), AG05252 (San Diego Zoological Society), KB3781 (San Diego Zoological Society), Jambi (Oregon Health and Science University) |
|---|---|
| Authentication | Each cell line was authenticated with karyotyping. We authenticated species for orangutans as described in Note S1 and and subspecies for chimpanzee as described in Note S2 |
| Mycoplasma contamination | The cell lines were not tested for mycoplasma contamination in our laboratories, but they were most likely tested for such contamination by cell line providers |
| Commonly misidentified lines<br>(See ICLAC register) | No commonly misidentified cell lines were used. |

## Plants

| Seed stocks | N/A: Plants were not part of this study |
|---|---|
| Novel plant genotypes | *Describe the methods by which all novel plant genotypes were produced. This includes those generated by transgenic approaches, gene editing, chemical/radiation-based mutagenesis and hybridization. For transgenic lines, describe the transformation method, the number of independent lines analyzed and the generation upon which experiments were performed. For gene-edited lines, describe the editor used, the endogenous sequence targeted for editing, the targeting guide RNA sequence (if applicable) and how the editor was applied.* |
| Authentication | *Describe any authentication procedures for each seed stock used or novel genotype generated. Describe any experiments used to assess the effect of a mutation and, where applicable, how potential secondary effects (e.g. second site T-DNA insertions, mosiacism, off-target gene editing) were examined.* |

