## [Peer Review File · Nature]

Manuscript Title: The Complete Sequence and Comparative Analysis of Ape Sex Chromosomes

Reviewer Comments & Author Rebuttals

Reviewer Reports on the Initial Version:

Referees' comments:

Referee #1 (Remarks to the Author):

This paper reports new T2T assemblies of the X- and Y-chromosomes of nearly all extant species of great apes, and includes the siamang as an outgroup. The assemblies of these chromosomes are extensively documented and analyzed. While high quality assemblies of great ape autosomal genomes are already available, the sex chromosomes have (as indicated by these authors) been seriously under-studied, for various reasons. Full length assemblies of these nonhuman primate sex chromosomes provide novel information about the long-term pattern of sex chromosome evolution and the mechanisms of those chromosomal changes. The work also places the previous T2T assemblies of the human X- and Y-chromosome into a broader comparative context.

The methods used to produce these assemblies are the state-of-the-art. I find no weaknesses and have no concerns regarding the quality of the assemblies. The figures included present significant aspects of the results. The authors have examined a wide range of issues, including but not limited to the rate of evolution in single copy and repetitive sequences, the composition of each sex chromosome in each species (i.e. protein coding genes, ampliconic regions, palindromic sequences, transposable elements and other features). For the first time, we have detailed evidence for the likely mechanisms of sex chromosome evolution as well as the organization of the centromeres and satellite DNA.

A number of major new results are presented. The authors map the pseudoautosomal region 2 in humans and bonobos, and report that these are the only species with PAR2. The results help to document and explain the rapid changes in the Y-chromosomes of multiple ape species, not just chimpanzees as previously reported. The identification of non-B DNA structures is significant, as are the inferences regarding the effects of purifying selection on these chromosomes, especially the Y. Overall, this work is a substantive, extensive and constitutes a unique contribution to comparative primate genomics.

However, I find the presentation of this work to be unnecessarily long. I think the manuscript can be made more concise and thus more effective if specific sections were reduced, possibly by moving some of the details to notes or supplementary text. One particular section that could be reduced in the main text without major loss of scientific impact is the section on the details of palindromic sequences (lines 249-269). The following section on repetitive elements (lines 277-312) could also be reduced.

Referee #2 (Remarks to the Author):

The authors report gapless, telomere-to-telomere assemblies of five great apes and a lesser ape. This impressive feat of Y chromosome assembly will be of broad interest to the scientific community and sets a new standard for assembling gapless sex chromosomes that future researchers can strive for. With this data, the authors revealed an exciting degree of lineage-specific structural evolution of Y chromosomes among fully resolved centromeric and ampliconic regions. I thought the methods used in this study were sound and the manuscript was clearly written and would be accessible to a broad audience. I only have suggestions on some areas of the manuscript that were not fully clear or suggestions for additional analyses or comparisons with existing studies.

Line 156: "In all cases the sex chromosomes were fully separate from the autosomes by Verkko, and several X and Y chromosomes were determined to be complete." How do the authors know that this method resolved these chromosomes 100%? Is it possible that some HiFi reads that share sequence homology between chromosomes were misassembled? I agree that this method likely resolved a majority or nearly all of the chromosomes, but how can the authors be confident that they were fully resolved?

Line 187: "... the Y chromosome experienced a greater degree of sequence turnover. . ." Do the authors mean lineage-specific sequence degeneration? Turnover here is unclear.

Line 215: "... consistent with male mutation bias." How do the authors know this is male mutation bias? A higher substitution rate on the Y chromosome relative to the X chromosome could also be caused by the smaller effective population size of the Y (the Y is 1/3 the effective population size of the X). Nearly neutral mutations fix at a higher rate in smaller populations, which would also cause this

pattern. Do the authors have data to distinguish these two possibilities? The authors should discuss that these higher rates could also be caused by differences in effective population size.

Line 432: "While X-degenerate Y-linked gene content..." I find this phrase a little confusing. This category of genes in other papers has been called "Ancestral single-copy." This would be more consistent with previous literature.

Figure 6B: The authors found single-copy ancestral genes that were independently retained across multiple lineages. This type of analysis has been conducted across other mammalian Y chromosomes, including the chimpanzee (e.g. see Bellott et al. 2014. Mammalian Y chromosomes retain widely expressed dosage-sensitive regulators). The authors cite this paper, but I don't see a comparison with the results of this paper. Are these the same retained core ancestral, single-copy genes that are also found among other mammalian lineages?

Line 395: In the section "Protein-coding gene evolution" the authors report positive and purifying selection across different categories of genes on the Y chromosome. Within the paper, the authors also overlay intraspecific population sequencing to study nucleotide diversity across the sex chromosomes. It would be useful to also use this data to compute McDonald-Kreitman statistics to verify if the population polymorphism data also supports whether different coding regions are under purifying or positive selection.

Referee #3 (Remarks to the Author):

The manuscript by Makova et al. on the chromosome structure and comparative sequence analysis of T2T ape sex chromosome assemblies sets the standard for comparative genomics in the era of complete genomes. The authors describe evolutionary genomic contrasts between the X and Y chromosomes regarding their structural divergence in humans and six apes. The authors identify remarkable stability on the X chromosome in comparison to the radically reshuffled Y chromosome. Although these aspects were hinted at in previous studies of ape Y and X chromosome comparisons, the present authors leave no stone unturned, now possessing the complete picture of complex divergence, including previously hidden satellites, segmental duplications, and centromeres. There is little to fault with the quality and rigor of the sequencing assembly, annotation, and analysis by this group of genome experts. Most of my

comments are for improving the readability and accessibility of this exciting yet dense science package to a broader audience.

1. Please clarify the terminology of the different repetitive sequences and gene classes: ampliconic regions are, by definition, segmental duplications...perhaps these definitions should be explained in the introduction. Even the distinction between multicopy gene families and ampliconic gene families could use clarity as important gene family changes may have occurred outside of ampliconic regions on the X and Y.

2. Using the term “novel sequences” in Figure 2 and others may confuse different readers. “Novel” has different meanings in the evolutionary genomics literature, including “a new gene or sequence” private to a lineage/species genome rather than the first to be sequenced for this species, which is what is intended here. I don’t think indicating an entire chromosome being “novel” (i.e., B. orangutan Y, siamang Y) is worth devoting to a single black data track. I suggest adding a new track for "lineage-specific" sequence and protein coding genes/gene family members as an alternative. This would help readers visualize the context of where and how clustered new and or recurrent sequence acquisitions have been.

3. In Figure 2, the Siamang Y is hard to find, tucked between the key and the X chromosomes. I recommend shifting the X chromosome panel down further and expanding horizontally so that apparent differences are clearer visually.

4. The evolutionary dynamics of ampliconic sequence gene gain and loss is one of the more exciting aspects of sex chromosome evolution. The manuscript focuses heavily on chrY divergence and less on X chromosome gene divergence, although the authors suggest there is far less on the latter, which was surprising. Indeed, there have been multicopy gene family expansions and losses on the X, too, although they may not all reside within amplicons. It would be nice to summarize these for the reader, too, just as was done in 6B for the Y, so that the treatment seems more balanced.

5. Following up on the previous point, I was somewhat disappointed by some of the display items in Figure 3. I appreciate there is a lot to communicate. Still, the Circos plots are too dense and confusing to do the results any justice and feel more like a data dump rather than an intuitive display and depiction of relevant changes between pairs of closely related species (chimp/bonobo or the two orangutans). I recommend reconsidering and reconfiguring these to include fewer and more specific examples, perhaps in a horizontal fashion. The gene names are cramped and difficult to read in the current format. To achieve this, I recommend moving the dot plots from A and B to the supplement, maybe even the whole Circos plots, and focusing on newer images illustrating novel gains and losses between select species pairs.

6. Extended Data Fig. 2. The orangutan haplotypes aren’t connected by a branch indicating distance. Was this intended?

7. Extended Data Fig. 3. A maximum likelihood phylogeny with appropriate model testing would be preferable. The branch lengths seem unusually variable for such a closely related species group and may have to do with how missing data was dealt with in GeneiousPrime. Please indicate how the phylogeny was rooted.
8. Methods: pseudogene identification: ...“where we identified genes with significant deviations (truncations) relative to their X chromosome counterparts as pseudogenes.” Please define what represents a significant deviation.
9. Fig S9, in the absence of coordinates, it would be helpful to indicate the p and q arms on the ideograms, either in the figure or the legend.
10. Fig. S10: it seems unusual for T2T assemblies that there are unknown repeat sequence classes in humans but not in apes. Can this be discussed and clarified somewhere for the reader?
11. Fig. S16. The labeling and explanation for panel C figures could use additional detail. What is the percentage (y-axis), and what is the number of samples on the x-axis?
12. Please check that the cited divergence dates among apes and humans are consistent with recent literature. The cited values are often derived from much older studies based on very small molecular and constrained gene-centric datasets. They seem on the younger side and may not be consistent with estimates from more expansive genomic sampling with multiple fossil calibrations, such as Vanderpool et al. 2020 (PLOS Biology, 18: e3000954), Álvarez-Carretero et al. 2022 (Nature, 602:263-267), Foley et al. 2023 (Science 380: eabl8189), Shao et al. 2023 (Science, 380: 913-924).

Author Rebuttals to Initial Comments:

Responses to Reviewers

Reviewer 1

Reviewer 1 (expertise in primate genomics) provided the following general comments that included a few recommendations regarding conciseness:

This paper reports new T2T assemblies of the X- and Y-chromosomes of nearly all extant species of great apes, and includes the siamang as an outgroup. The assemblies of these chromosomes are extensively documented and analyzed. While high quality assemblies of great ape autosomal genomes are already available, the sex chromosomes have (as indicated by these authors) been seriously under-studied, for various reasons. Full length assemblies of these nonhuman primate sex chromosomes provide novel information about the long-term pattern of sex chromosome evolution and the mechanisms of those chromosomal changes. The work also places the previous T2T assemblies of the human X- and Y-chromosome into a broader comparative context.

The methods used to produce these assemblies are the state-of-the-art. I find no weaknesses and have no concerns regarding the quality of the assemblies. The figures included present significant aspects of the results. The authors have examined a wide range of issues, including but not limited to the rate of evolution in single copy and repetitive sequences, the composition of each sex chromosome in each species (i.e. protein coding genes, ampliconic regions, palindromic sequences, transposable elements and other features). For the first time, we have detailed evidence for the likely mechanisms of sex chromosome evolution as well as the organization of the centromeres and satellite DNA.

A number of major new results are presented. The authors map the pseudoautosomal region 2 in humans and bonobos, and report that these are the only species with PAR2. The results help to document and explain the rapid changes in the Y-chromosomes of multiple ape species, not just chimpanzees as previously reported. The identification of non-B DNA structures is significant, as are the inferences regarding the effects of purifying selection on these chromosomes, especially the Y. Overall, this work is a substantive, extensive and constitutes a unique contribution to comparative primate genomics.

However, I find the presentation of this work to be unnecessarily long. I think the manuscript can be made more concise and thus more effective if specific sections were reduced, possibly by moving some of the details to notes or supplementary text. One particular section that could be reduced in the main text without major loss of scientific impact is the section on the details of palindromic sequences (lines 249-269). The following section on repetitive elements (lines 277-312) could also be reduced.

We appreciate this reviewer's positive evaluation of our manuscript. We have made the manuscript more concise overall while paying particular attention to the suggested sections.

Item 1

Suggestion: One particular section that could be reduced in the main text without major loss of scientific impact is the section on the details of palindromic sequences (lines 249-269).

Response: This section was reduced by 69% (from 729 to 211 words).

Item 2

Suggestion: The following section on repetitive elements (lines 277-312) could also be reduced.

Response: This section was also reduced by 30% (from 607 to 425 words).

Reviewer 2

Reviewer 2 (expertise in sex chromosome evolution) provided the following general comments:

The authors report gapless, telomere-to-telomere assemblies of five great apes and a lesser ape. This impressive feat of Y chromosome assembly will be of broad interest to the scientific community and sets a new standard for assembling gapless sex chromosomes that future researchers can strive for. With this data, the authors revealed an exciting degree of lineage-specific structural evolution of Y chromosomes among fully resolved centromeric and ampliconic regions. I thought the methods used in this study were sound and the manuscript was clearly written and would be accessible to a broad audience. I only have suggestions on some areas of the manuscript that were not fully clear or suggestions for additional analyses or comparisons with existing studies.

We are grateful to this reviewer for their time and suggestions. We list each of the specific comments below and address them in turn.

Item 1

Question: Line 156: "In all cases the sex chromosomes were fully separate from the autosomes by Verkko, and several X and Y chromosomes were determined to be complete." How do the authors know that this method resolved these chromosomes 100%? Is it possible that some HiFi reads that share sequence homology between chromosomes were misassembled? I agree that this method likely resolved a majority or nearly all of the chromosomes, but how can the authors be confident that they were fully resolved?

Response: We are confident that the sex chromosomes have been assembled in their entirety because of (1) the presence of telomeric arrays on both ends of each assembly, (2) the lack of any assembly graph edges between sex and autosomal components, and (3) the uniformity of sequencing read coverage mapped across the entire length of the assemblies. (1) Demonstrates that we have reached the "end" of each chromosome, and the location of these telomeric repeats is consistent with prior knowledge, e.g. in terms of their relative location to the pseudoautosomal regions (PARs). (2) Indicates a lack of significant sequence similarity between the sex chromosomes and the autosomes. There is certainly some homology between shared repeats on these chromosomes (e.g. *Alus*, LINEs, etc.), but the vast majority of them are relatively short (e.g. <10 kb) and diverged (e.g. <99%). The absence of edges linking the sex and autosomal components of the graph tells us that these repeats were easily distinguished with the length and

accuracy of the sequencing technologies used. (3) Follows standard assembly validation practices and confirms the structural integrity of the assemblies. Stretches of missing sequence would result in clipped read alignments and a drop in mapped coverage at the deletion site, while stretches of collapsed sequence would result in artificially high coverage. Instead, we observed uniformly mapped reads consistent with a random sampling process (albeit with some differences caused by sequencing bias in certain contexts). This is the typical argument made by Bayesian assembly validation approaches. We added a short note to the Methods summarizing this:

“Validation of the assemblies was done in multiple ways to assess assembly completeness and correctness. Coverage analysis, erroneous k -mers, and haplotype-specific k -mers (for the two trios) were manually inspected using IGV, and assembly QV was calculated using Merqury. Completeness of each chromosome was confirmed by the identification of telomeric arrays on each end and uniform coverage of long read mappings, with an absence of clipped reads or other observable mapping artifacts.”

Item 2

Question: Line 187: “... the Y chromosome experienced a greater degree of sequence turnover...” Do the authors mean lineage-specific sequence degeneration? Turnover here is unclear.

Response: We have changed this sentence to “Taken together, these observations suggest greater interspecific differences on the Y than on the X”.

Item 3

Question: Line 215: “... consistent with male mutation bias.” How do the authors know this is male mutation bias? A higher substitution rate on the Y chromosome relative to the X chromosome could also be caused by the smaller effective population size of the Y (the Y is 1/3 the effective population size of the X). Nearly neutral mutations fix at a higher rate in smaller populations, which would also cause this pattern. Do the authors have data to distinguish these two possibilities? The authors should discuss that these higher rates could also be caused by differences in effective population size.

Response: We agree with the reviewer that the smaller effective population size of the Y could in principle lead to a higher substitution rate of nearly neutral mutations. However, below we argue that this effect would contribute only minimally to the observed two-fold higher substitution rate on the Y compared to the X. First, comparative genomic studies suggest that only 8% of the human genome is subject to negative selection¹, and similar estimates have been derived for the Y chromosome². Second, about 50% of effectively selected mutations are strongly deleterious ($|s| > 0.001$) and thus fix neither on the X nor the Y³. Therefore, under the conservative assumption that nearly neutral mutations do not fix on the X chromosome but fix neutrally on the Y chromosome, this effect explains at most a 4% higher divergence on the Y compared to the X. This is negligible compared to the observed 176% higher human-chimpanzee divergence on the Y compared to the X. Thus, another explanation, i.e. male mutation bias, should be evoked. At least a two-fold male mutation bias is broadly observed across mammalian species when estimated from parent-offspring trios, further supporting our mutational explanation⁴.

Item 4

Comment: Line 432: “While X-degenerate Y-linked gene content...” I find this phrase a little confusing. This category of genes in other papers has been called “Ancestral single-copy.” This would be more consistent with previous literature.

Response: In the primate Y chromosome literature two terms for these genes were used: “X-degenerate”⁵⁻⁷ and “ancestral”⁸. Because the term “ancestral” can also be used for homologous (gametologous) genes on the X chromosome, and to save space, we have changed the naming of these genes on both X and Y to “ancestral”. We mention that on the Y these are the same genes as “X-degenerate”.

Item 5

Question: Figure 6B: The authors found single-copy ancestral genes that were independently retained across multiple lineages. This type of analysis has been conducted across other mammalian Y chromosomes, including the chimpanzee (e.g. see Bellott et al. 2014. Mammalian Y chromosomes retain widely expressed dosage-sensitive regulators). The authors cite this paper, but I don’t see a comparison with the results of this paper. Are these the same retained core ancestral, single-copy genes that are also found among other mammalian lineages?

Response: Thank you for this comment. We have now performed a comparison of our results with the results of Bellott et al. 2014 and have added the following sentence to the paper: “Notably, all four ancestral single-copy genes found to be retained in eutherian mammals in another study⁹ were present in the primates studied, and three of them (*DDX3Y*, *UTY*, and *ZFY* but not *SRY*) exhibited signatures of purifying selection.” Additionally, we have performed a detailed comparison of single-copy chrY ancestral genes’ presence/absence/pseudogenization resulting from our T2T ape genome gene annotations vs. several previous publications. These are presented in Table S39. With a handful of exceptions, the results of our study were remarkably concordant with those of previous studies.

Item 6

Suggestion: Line 395: In the section “Protein-coding gene evolution” the authors report positive and purifying selection across different categories of genes on the Y chromosome. Within the paper, the authors also overlay intraspecific population sequencing to study nucleotide diversity across the sex chromosomes. It would be useful to also use this data to compute McDonald-Kreitman statistics to verify if the population polymorphism data also supports whether different coding regions are under purifying or positive selection.

Response: We have performed this analysis, and its results are presented in Note S13. The results were not significant, likely due to the lack of power.

Reviewer 3

Reviewer 3 (expertise in mammalian genomics) provided the following general comments:

The manuscript by Makova et al. on the chromosome structure and comparative sequence analysis of T2T ape sex chromosome assemblies sets the standard for comparative genomics in the era of complete genomes. The authors describe evolutionary genomic contrasts between the X and Y chromosomes regarding their structural divergence in humans and six apes. The authors identify remarkable stability on the X chromosome in comparison to the radically reshuffled Y chromosome. Although these aspects were hinted at in previous studies of ape Y and X chromosome comparisons, the present authors leave no stone unturned, now possessing the complete picture of complex divergence, including previously hidden satellites, segmental duplications, and centromeres. There is little to fault with the quality and rigor of the sequencing assembly, annotation, and analysis by this group of genome experts. Most of my comments are for improving the readability and accessibility of this exciting yet dense science package to a broader audience.

We appreciate this reviewer's constructive criticism and are grateful for their time and help. We list each of the specific comments and address each of them below.

Item 1

Suggestion: Please clarify the terminology of the different repetitive sequences and gene classes: ampliconic regions are, by definition, segmental duplications...perhaps these definitions should be explained in the introduction. Even the distinction between multicopy gene families and ampliconic gene families could use clarity as important gene family changes may have occurred outside of ampliconic regions on the X and Y.

Response: We appreciate this comment. While certainly related, historically, ampliconic regions on the Y chromosome and segmental duplications have been defined differently. Per Skaletsky *et al.*⁶, ampliconic regions are long multi-copy regions with >50% sequence identity. Per Bailey *et al.* 2001¹⁰, segmental duplications are >1-kb multi-copy regions with >90% identity. We now define them in detail in the main text and Methods, and have included Note S5 describing an overlap and differences between these two groups of regions, as well as palindromes.

We have also clarified the nomenclature used to define multi-copy and ampliconic gene families. We have searched for multi-copy gene families using criteria used in Assis & Bachtrog¹¹. Namely, to be called multi-copy (i.e., duplicate) genes, we required copies to have at least 50% protein sequence identity according to BLAST over >35% of the length of each sequence. Among multi-copy gene families, we defined ampliconic genes as those with $\geq 97\%$ sequence identity, as this is consistent with other studies (e.g., Zhou *et al.* 2023⁷) and is supported by the natural breakpoint in sequence identity in our data (shown below for pairwise protein sequence identities for the Y chromosome multi-copy genes, Fig. S20):

We provide annotations of both multi-copy and ampliconic genes on the X and the Y (Table S36 and Table S37), however we focus on ampliconic genes as they are frequently located in palindromes and thus represent the hallmarks of sex chromosomes.

Item 2

Suggestion: Using the term “novel sequences” in Figure 2 and others may confuse different readers. “Novel” has different meanings in the evolutionary genomics literature, including “a new gene or sequence” private to a lineage/species genome rather than the first to be sequenced for this species, which is what is intended here. I don’t think indicating an entire chromosome being “novel” (i.e., B. orangutan Y, siamang Y) is worth devoting to a single black data track. I suggest adding a new track for “lineage-specific” sequence and protein coding genes/gene family members as an alternative. This would help readers visualize the context of where and how clustered new and or recurrent sequence acquisitions have been.

Response: Thank you for this comment. Our motivation behind having the track with the sequences presented here for the first time was two-fold: (1) to highlight the new sequencing data this study adds as compared to the previous literature (this is where black rectangles are important); (2) to provide a visualization of a correlation between

such regions and regions enriched in non-B DNA. We have changed the name of this track to “New” and explained what we present in this track in the legend.

Regarding the second suggestion, we have added the gene density track to Fig. 2, and have presented the lineage-specific repeat track in Ext. Data Fig. 2. A separate track of lineage-specific sequences is possible to produce but will heavily depend on the phylogenetic distances between the studied species, and thus we have decided not to include it.

Item 3

Suggestion: In Figure 2, the Siamang Y is hard to find, tucked between the key and the X chromosomes. I recommend shifting the X chromosome panel down further and expanding horizontally so that apparent differences are clearer visually.

Response: We have reorganized this figure to make the siamang Y chromosome easier to find.

Item 4

Suggestion: The evolutionary dynamics of ampliconic sequence gene gain and loss is one of the more exciting aspects of sex chromosome evolution. The manuscript focuses heavily on chrY divergence and less on X chromosome gene divergence, although the authors suggest there is far less on the latter, which was surprising. Indeed, there have been multicopy gene family expansions and losses on the X, too, although they may not all reside within amplicons. It would be nice to summarize these for the reader, too, just as was done in 6B for the Y, so that the treatment seems more balanced.

Response: We appreciate this suggestion. We have analyzed multi-copy genes on the X chromosome, their location among sequence classes, and identified ampliconic gene families among them (Table S37). Although our focus has been on the Y chromosome, we have included the following summary of our results of the analysis of the X chromosome: “Among multi-copy genes on the Y and on the X, we detected ampliconic gene families, defined as families with at least two copies having $\geq 97\%$ sequence identity at the protein level in at least one species (Table S36, Table S37). Many of them were located in palindromes. The proportion of ampliconic among multi-copy gene families was lower on the X than the Y chromosome (29/123 vs. 14/20, $p=2.68 \times 10^{-6}$, chi-squared test) highlighting a lower percentage of the X length covered by ampliconic sequences and palindromes (Table S14). Nevertheless, we still found several abundant ampliconic gene families on the X—*GAGE*, *MAGE*, and *SPANX*, the products of which are exclusively or predominantly expressed in testis (Table S37).”

Item 5

Suggestion: Following up on the previous point, I was somewhat disappointed by some of the display items in Figure 3. I appreciate there is a lot to communicate. Still, the Circos plots are too dense and confusing to do the results any justice and feel more like a data dump rather than an intuitive display and depiction of relevant changes between pairs of closely related species (chimp/bonobo or the two orangutans). I recommend reconsidering and reconfiguring these to include fewer and more specific examples, perhaps in a horizontal fashion. The gene names are cramped and difficult to read in the current format. To achieve this, I recommend moving the dot plots from A and B to the

supplement, maybe even the whole Circos plots, and focusing on newer images illustrating novel gains and losses between select species pairs.

Response: We agree that our presentation of these results could have been improved. To do this, we have reorganized this figure (Fig. 3) completely. We now provide a horizontal presentation of palindromes and their sharing, highlight a few relevant gene family expansion examples, and conclude this figure by a comparison of gene density across different sequence classes on the X and the Y. We moved dot plots to Ext. Data Fig. 1. Instead of presenting the circus plots, we provide an interactive version of the horizontal representation of palindrome sharing which allows one to identify how the genes are shared (<https://observablehq.com/d/6e3e88a3e017ec21>).

Item 6

Question: Extended Data Fig. 2. The orangutan haplotypes aren't connected by a branch indicating distance. Was this intended?

Response: We have modified this figure (now Ext. Data Fig. 4) as requested.

Item 7

Suggestion: Extended Data Fig. 3. A maximum likelihood phylogeny with appropriate model testing would be preferable. The branch lengths seem unusually variable for such a closely related species group and may have to do with how missing data was dealt with in GeneiousPrime. Please indicate how the phylogeny was rooted.

Response: Based on the suggestion by the Reviewer, we have reanalyzed these data using a Maximum Likelihood method. Our results (presented in Ext. Data Fig. 7) are broadly consistent with our previous results. We have included the following in the Methods:

“TSPY gene analysis. The UCSC table browser was used to retrieve and export the TSPY sequences. For every genome, the appropriate gene annotation dataset was selected with the specific regions defined using the locations of the curated TSPY copies. The sequence was retrieved in the 5' UTR, CDS exons, 3' UTR, and intron regions and the generated fasta files were then used for alignment with MAFFT (v7.520)¹². Maximum-likelihood phylogenies were inferred using IQTree (v2.0.3)¹³ with the best-fit substitution model estimated by ModelFinder¹⁴ (best-fit model according to BIC: TVM+F+G4). Node support values were estimated using 10,000 ultrafast bootstrap replicates¹⁵ with hill-climbing nearest neighbor interchange (`-bnni` flag) to avoid severe model violations. Nodes with <95% ultrafast bootstrap support were collapsed as polytomies.”

Item 8

Suggestion: Methods: pseudogene identification: ...“where we identified genes with significant deviations (truncations) relative to their X chromosome counterparts as pseudogenes.” Please define what represents a significant deviation.

Response: We have included additional details by modifying the text to “All present genes were aligned to their orthologs and their gametologs, where we identified genes with significant deviations (truncations of 20% or greater) relative to known (functional) Y copies in other ape species, or their X chromosome counterpart, as pseudogenes (Table S39).” In addition, we have provided additional clarifying information in Table S39.

Item 9

Suggestion: Fig. S9, in the absence of coordinates, it would be helpful to indicate the p and q arms on the ideograms, either in the figure or the legend.

Response: We have modified this figure to show centromeres.

Item 10

Question: Fig. S10: it seems unusual for T2T assemblies that there are unknown repeat sequence classes in humans but not in apes. Can this be discussed and clarified somewhere for the reader?

Response: We thank the reviewer for pointing out this discrepancy in repeats classified as “Unknown,” which was caused by the addition of newly discovered repeat models in the CHM13 repeat library. We have made adjustments to Figure S18 and corresponding Table S20 to appropriately sort repeat types erroneously classified as “Unknown” in CHM13 into their respective subtypes.

Item 11

Question: Fig. S16. The labeling and explanation for panel C figures could use additional detail. What is the percentage (y-axis), and what is the number of samples on the x-axis?

Response: We have revised the legend for panel C to improve clarity (now it is Fig. S17). The updated legend is as follows: “Allele frequency histogram of variants across shared subspecies samples relative to the T2T and previous genome assemblies. The x-axis indicates the number of subspecies samples, and the y-axis represents the percentage of variants observed within these samples.”

Item 12

Suggestion: Please check that the cited divergence dates among apes and humans are consistent with recent literature. The cited values are often derived from much older studies based on very small molecular and constrained gene-centric datasets. They seem on the younger side and may not be consistent with estimates from more expansive genomic sampling with multiple fossil calibrations, such as Vanderpool et al. 2020 (PLOS Biology, 18: e3000954), Álvarez-Carretero et al. 2022 (Nature, 602:263-267), Foley et al. 2023 (Science 380: eabl8189), Shao et al. 2023 (Science, 380: 913-924).

Response: We appreciate this comment. We have included additional references for divergence times (Table S1) and have revised them in Fig. 1A. These divergence times are now computed as the averages of estimates from multiple studies, as shown in detail

in Table S1. We have included the references we used previously, as well as all but one suggested by the reviewer. Indeed, we believe that the divergence time estimates in Álvarez-Carretero et al. 2022 (Nature, 602:263-267) have very high upper bounds (e.g., divergence with Old World monkeys was provided as an upper bound for human-gorilla divergence). The values in our manuscript still fall within the confidence intervals suggested by that study.

References

1. Rands, C. M., Meader, S., Ponting, C. P. & Lunter, G. 8.2% of the Human genome is constrained: variation in rates of turnover across functional element classes in the human lineage. *PLoS Genet.* **10**, e1004525 (2014).
2. Wilson Sayres, M. A., Lohmueller, K. E. & Nielsen, R. Natural selection reduced diversity on human y chromosomes. *PLoS Genet.* **10**, e1004064 (2014).
3. Kim, B. Y., Huber, C. D. & Lohmueller, K. E. Inference of the Distribution of Selection Coefficients for New Nonsynonymous Mutations Using Large Samples. *Genetics* **206**, 345–361 (2017).
4. Bergeron, L. A. *et al.* Evolution of the germline mutation rate across vertebrates. *Nature* **615**, 285–291 (2023).
5. Hughes, J. F. *et al.* Chimpanzee and human Y chromosomes are remarkably divergent in structure and gene content. *Nature* **463**, 536–539 (2010).
6. Skaletsky, H. *et al.* The male-specific region of the human Y chromosome is a mosaic of discrete sequence classes. *Nature* **423**, 825–837 (2003).
7. Zhou, Y. *et al.* Eighty million years of rapid evolution of the primate Y chromosome. *Nat Ecol Evol* **7**, 1114–1130 (2023).
8. Hughes, J. F. *et al.* Strict evolutionary conservation followed rapid gene loss on human and rhesus Y chromosomes. *Nature* **483**, 82–86 (2012).
9. Bellott, D. W. *et al.* Mammalian Y chromosomes retain widely expressed dosage-sensitive regulators. *Nature* **508**, 494–499 (2014).

10. Bailey, J. A., Yavor, A. M., Massa, H. F., Trask, B. J. & Eichler, E. E. Segmental duplications: organization and impact within the current human genome project assembly. *Genome Res.* **11**, 1005–1017 (2001).
11. Assis, R. & Bachtrög, D. Neofunctionalization of young duplicate genes in *Drosophila*. *Proc. Natl. Acad. Sci. U. S. A.* **110**, 17409–17414 (2013).
12. Katoh, K. & Standley, D. M. MAFFT multiple sequence alignment software version 7: improvements in performance and usability. *Mol. Biol. Evol.* **30**, 772–780 (2013).
13. Minh, B. Q. *et al.* IQ-TREE 2: New Models and Efficient Methods for Phylogenetic Inference in the Genomic Era. *Mol. Biol. Evol.* **37**, 1530–1534 (2020).
14. Kalyaanamoorthy, S., Minh, B. Q., Wong, T. K. F., von Haeseler, A. & Jermiin, L. S. ModelFinder: fast model selection for accurate phylogenetic estimates. *Nat. Methods* **14**, 587–589 (2017).
15. Hoang, D. T., Chernomor, O., von Haeseler, A., Minh, B. Q. & Vinh, L. S. UFBoot2: Improving the Ultrafast Bootstrap Approximation. *Mol. Biol. Evol.* **35**, 518–522 (2018).

Reviewer Reports on the First Revision:

Referees' comments:

Referee #2 (Remarks to the Author):

I appreciate the thorough responses to my comments. The authors have addressed everything. I think this exciting study is now suitable for publication. Great work!

Referee #2 (Remarks on code availability):

I was able to access the code through github. I found the readme files appropriately commented. To this extent, the code seems accessible and a useful resource for the community. I did not evaluate the details of the code.

Referee #3 (Remarks to the Author):

The authors have done an excellent job addressing the points raised in my prior review. I have no further comments.